# Water vapour isotopes over West Africa as observed from space: which processes control tropospheric $H_2O$/HDO pair distributions?

Christopher J. Diekmann[1,a], Matthias Schneider[1], Peter Knippertz[2], Tim Trent[4,5], Hartmut Boesch[4,c], Amelie Ninja Roehling[1], John Worden[6], Benjamin Ertl[1,3], Farahnaz Khosrawi[1,b], and Frank Hase[1]

[1]Institute of Meteorology and Climate Research, Atmospheric Trace Gases and Remote Sensing, Karlsruhe Institute of Technology, Karlsruhe, Germany
[2]Institute of Meteorology and Climate Research, Troposphere Research, Karlsruhe Institute of Technology, Karlsruhe, Germany
[3]Scientific Computing Center, Karlsruhe Institute of Technology, Karlsruhe, Germany
[4]Earth Observation Science, Department of Physics and Astronomy, University of Leicester, Leicester, UK
[5]National Centre for Earth Observation, Department of Physics and Astronomy, University of Leicester, Leicester, UK
[6]Jet Propulsion Laboratory, California Institute of Technology, Pasadena, California, USA
[a]now at: Remote Sensing Products Division, EUMETSAT, Darmstadt, Germany
[b]now at: Jülich Supercomputing Centre, Forschungszentrum Jülich GmbH, Jülich, Germany
[c]now at: Institute of Environmental Physics, University of Bremen, Bremen, Germany

**Correspondence:** Christopher J. Diekmann (chjdiekmann@gmail.com)

**Abstract.** The West African Monsoon (WAM) represents the main source of rainfall over West Africa and thus has important socio-economic impacts. However, the complex interactions of the large-scale circulation, convective dynamics and microphysical processes associated with the WAM make it difficult to disentangle the contributions of individual atmospheric processes to the hydrological cycle in observations and models.

Making use of recent advances in retrieving the isotopic composition of tropospheric water vapour from space, we promote the paired analysis of $H_2O$ and HDO to investigate the moisture pathways and processes associated with the WAM. Data from the state-of-the-art satellite sensors IASI, AIRS and TROPOMI, together with the multi-satellite IMERG precipitation product, serve to characterize the variability of $H_2O$ and HDO (with their ratio product $\delta D$) over West Africa at the convective and seasonal scales and with respect to impacts from dynamical and microphysical processes. In particular, we find: (1) Monsoon convection over the Sahel leads to a marked anti-correlation between increasing $H_2O$ and decreasing $\delta D$ in the free troposphere. This is due to dry air intrusions from the Saharan upper troposphere that feed into Sahelian squall line systems, foster rain evaporation and, hence, lead to $\delta D$ depletion in the mid-tropospheric vapor; (2) Over the Guinea Coast, convective precipitation is associated with overall moist and enriched signals . Here, surface evaporation from the Tropical Atlantic moistens the troposphere, reducing the efficiency of the rain evaporation and the corresponding $\delta D$ depletion; (3) During the Sahelian monsoon peak, an anti-correlation between increasing precipitation amount and decreasing $\delta D$ becomes apparent in a year-to-year comparison. Thus, this provides observational evidence for the *amount effect* in tropospheric water vapour, similar to what is known for the isotopic composition in precipitation; (4) When no considerable precipitation occurs, e.g. during the Sahelian winter, the {$H_2O$, $\delta D$} signals point to mixing of dry air masses from different origins.

This study is the first to apply the new and comprehensive isotopic datasets of IASI, TROPOMI and AIRS to the WAM. Despite

the large complexity of the monsoon system, it confirms that satellite-based $H_2O$ and $\delta D$ data are useful to detect the impact of different microphysical and dynamical processes on the isotopic composition of water vapour at the seasonal and convective scales.

## 1 Introduction

The West African Monsoon (WAM) is one of the most prominent atmospheric features of the West African climate system (Fink et al., 2017). Over the semi-arid Sahel zone, the WAM causes a pronounced annual cycle that ranges from very dry winter months, when northeasterly Harmattan winds from the Sahara dominate, to a marked rainy season during summer, when southwesterly monsoon winds transport moisture from the Tropical Atlantic into the Sahel and feed intense convective rainfalls, often related to westward propagating squall lines (Lafore et al., 2017). With the WAM thus being the main source of annual precipitation over the Sahel (Dhonneur, 1981; Fink et al., 2006), its socio-economic impacts are enormous, as the region heavily relies on rain-fed agriculture (Sultan et al., 2005; Berntell et al., 2018). From a climatological perspective, changes in the WAM system are critical for West Africa, e.g. for the contribution to the greenhouse effect (Spencer and Braswell, 1997) and for the degree of desiccation, i.e. the progressive soil drying and subsequent desertification as a result of decreasing rainfall and intensifying drought (Hulme, 2001). In addition, the WAM system is also found to affect weather systems in remote areas through dynamical teleconnections, such as over the North Atlantic and Europe (Bielli et al., 2010; Gaetani et al., 2011; Pante and Knippertz, 2019).

The simulation of the atmospheric processes related to the WAM, however, still poses a major challenge for modern numerical weather and climate models (Roehrig et al., 2013; Vogel et al., 2018). This is mainly due to the strong non-linearity in the coupling of atmospheric circulation, radiative transfer, land surface features, boundary layer processes as well as microphysical processes associated with convection (e.g. Colman et al., 2017; Lafore et al., 2017). Even a seemingly accurate precipitation forecast may be a result of error compensation within the model, with the effect of potentially reducing the robustness of climate projections using this model. Studies have shown that, besides other factors such as changes in Atlantic sea surface temperatures (e.g. Monerie et al., 2023) one of the main sources of uncertainty in climate projections over the Sahel is convection, mostly due to insufficient spatio-temporal resolution and the poor simulation of the convective organization and interaction with its environment (Marsham et al., 2013; de Vries et al., 2022). In particular convective mixing processes play a crucial role in controlling the free tropospheric water vapour budget, which in turn affects the radiative transfer and atmospheric stability, but accurately simulating these processes can be particularly challenging (Sherwood et al., 2014). In addition, the network of observations is limited in West Africa (Parker et al., 2008) and may not be adequate to capture the effects of the controlling processes required to enhance our general process understanding and to improve model performance.

A promising approach to provide new insights into different branches of the hydrological cycle and to validate numerical models is to inspect the isotopic composition of water in the atmosphere, in all its forms: liquid, gaseous and solid. As each water isotopologue (in the following, referred to as water isotope) is associated with characteristic binding energies and diffusivities, the ratios of different isotope concentrations are altered during phase changes. In this way, paired distributions of the

light water isotope $H_2O$ against heavier molecules such as HDO can be used to track effects from microphysical and dynamical processes and thus reveal unique insights into the history of observed air masses (e.g. Worden et al., 2007; Noone et al., 2011; Noone, 2012; Bolot et al., 2013; Galewsky et al., 2016; González et al., 2016; Schneider et al., 2016; Lacour et al., 2018; Risi et al., 2021). Typically, the ratio between HDO and $H_2O$ is given in the $\delta D$ notation (in ‰) with respect to the standardized ratio $R_s = 3.1152 \times 10^{-4}$ (Craig, 1961):

$$\delta D = \left( \frac{HDO/H_2O}{R_s} - 1 \right) \cdot 1000, \tag{1}$$

and concomitant distributions of $H_2O$ and $\delta D$ are often referred to as {$H_2O$, $\delta D$} pairs.

Previous studies have investigated tropospheric distributions of stable water isotopes during the WAM using campaign-based observations of near-surface humidity (Risi et al., 2008b, 2010a; Tremoy et al., 2012, 2014), spatio-temporally limited retrievals from historical space-borne sensors (Risi et al., 2010b) and isotope-enabled atmospheric models (Risi et al., 2010a; Diekmann et al., 2021a; Risi et al., 2023). These studies underline the strong influence of large-scale dynamics and microphysical processes on the isotopic composition of rain and water vapour. In particular, partial evaporation of falling rain droplets in the boundary layer and free troposphere has been linked to convective systems such as squall lines, and was found to account for an increased depletion of $\delta D$ observed in ambient vapour. By applying a model-based Lagrangian process attribution Diekmann et al. (2021a) confirmed in a reverse engineering approach that the paired {$H_2O$, $\delta D$} phase space is particularly suited to represent effects of the main atmospheric processes controlling the WAM. They found that its isotopic signatures in the troposphere during the summer monsoon period are well-explained by means of theoretical formulations derived by Noone (2012): (1) dynamical processes, in particular the mixing of moist air masses from the Tropical Atlantic with dry air from the Sahara affected by large-scale subsidence, and (2) microphysical processes, namely condensation and evaporation equilibration of liquid hydrometeors. In order to evaluate these theoretical concepts with real observations, new tropospheric datasets of pairs of $H_2O$ and $\delta D$ with dense spatio-temporal coverage are needed, which, however, have only recently become available for West Africa.

Despite its smaller abundance compared to other heavy isotopes in atmospheric water vapour molecules, deuterium has distinct spectroscopic features in the infrared electromagnetic range, allowing detection and quantification from remote sensing sensors (Clerbaux et al., 2009). With recent advances in the development of retrieval processors for state-of-the-art satellite sensors in terms of computational efficiency and data storage, new spaceborne datasets of tropospheric {$H_2O$, $\delta D$} pairs with high spatio-temporal coverage over multiple years have become available. For instance, corresponding retrievals were performed for observations on daily and global basis from the current infrared sensors Metop/IASI (Infrared Atmospheric Sounding Interferometer; Schneider et al., 2012, 2022; Diekmann et al., 2021b), Aqua/AIRS (Atmospheric Infrared Sounder; Worden et al., 2019) and Sentinel-5P/TROPOMI (Tropospheric Monitoring Instrument; Schneider et al., 2020; Trent et al., 2021). The growing availability of such datasets creates new opportunities for investigating atmospheric moisture processes from convective to climatic scales.

In this study, we make use from the high spatio-temporal coverage given by the {$H_2O$, $\delta D$} pair datasets from IASI, AIRS and TROPOMI in order to investigate the isotopic signatures of tropospheric water vapour over West Africa. Building on an

**Table 1.** Remote sensing datasets used within this study, sampled for the region of interest over West Africa ($0° – 15°$ N, $8°$ W – $8°$ E).

| Platform | Data Product | Hor. Res. | Local Overpass | Data Availability |
|---|---|---|---|---|
| IASI | {$H_2O$, $\delta D$} profiles | 12 km | 09.30 and 21.30 | Jan 2015 – Dec 2020 |
| AIRS | {$H_2O$, $\delta D$} profiles | 13.5 km | 13.30 | Jan 2015 – Dec 2019 |
| TROPOMI | {$H_2O$, $\delta D$} total columns | 5.5 – 7 km | 13.30 | Apr 2018 – Dec 2020 |
| IMERG | daily mean prec. | 0.1° | - | Jun – Jul 2016, 2018 – 2020 |
| | monthly mean prec. | | | Jan 2018 – Dec 2020 |

observational basis, the aim is to shed new light on atmospheric processes governing the WAM development, in particular on microphysical processes associated to convection and on the impact from large-scale circulation. The focus will be on:

- effects of convection in the {$H_2O$, $\delta D$} phase space on a daily event basis,

– the interannual variability of convection-related {$H_2O$, $\delta D$} signals,

- the quasi-climatological seasonal cycle of {$H_2O$, $\delta D$} signals.

For this purpose, this study also includes the hourly precipitation rates from the multi-satellite precipitation product IMERG (Integrated Multi-Satellite Retrievals for GPM; Huffman et al., 2014) to link the observed {$H_2O$, $\delta D$} pair signals to the prevailing precipitation conditions. We further utilize the IMERG data to detect individual major convective events as well as

pronounced dry periods, with the aim to identify the corresponding {$H_2O$, $\delta D$} observations and inspect the isotopic signatures for the different conditions. In order to foster the process-based understanding of the observed {$H_2O$, $\delta D$} pair signals, we add the theoretical process curves from Diekmann et al. (2021a) to assess effects from rain-vapour interactions and large-scale circulation to the {$H_2O$, $\delta D$} phase space.

The paper is structured as follows: Section 2 documents the research datasets and analysis methods considered throughout

this study. We analyze the remote sensing data for different target regions over West Africa on the convective scale (Sect. 3), with respect to the year-to-year variability (Sect. 4) and with a focus on the mean seasonal cycle (Sect. 5). The main conclusions and an outlook are given in Sect. 6.

## 2   Data and Methods

With the aim to characterize and further understand the {$H_2O$, $\delta D$} pair variability over West Africa, this study makes use of

observational datasets from space-based remote sensing (see Table 1) in combination with additional interpretative frameworks (see Sect. 2.1.1 and see 2.2). Domains over the Guinea Coast and the Sahel serve as study regions with different climatological conditions and are defined in alignment with Knippertz et al. (2017), i.e. latitudinal bounds of $0°$ to $7.5°$ N and $7.5°$ to $15°$ N, respectively, with common longitudinal bounds of $8°$ W to $8°$ E. These domains are found to be useful for documenting the shift of maximum precipitation from the coastal to the Sahelian regions, which is a key criterion for defining the onset and

further evolution of the WAM (Fitzpatrick et al., 2015; Knippertz et al., 2017; Diekmann et al., 2021a).

## 2.1 Tropospheric water isotope products from remote sensing

### 2.1.1 Dataset description

The main data sources for this study are three long-term and global datasets of tropospheric $\{H_2O, \delta D\}$ observations that were
115 retrieved from state-of-the-art satellite sensors and made available in recent years (see Table 1). In the following we briefly
describe these products and highlight selected particularities that are relevant for this study. Due to the distinct spectroscopic
features of deuterium in the infrared spectrum, infrared sounders have been proven useful to retrieve HDO abundances in the
troposphere. However, water vapour products from these sensors are currently limited to cloud-free conditions, thus inducing
a dry bias in larger-scale water vapour distributions (Schneider et al., 2010).
The first water isotope product is the $\{H_2O, \delta D\}$ pair dataset generated from thermal infrared spectra from Metop/IASI (Diek-
mann et al., 2021b; Schneider et al., 2022), developed and published in the framework of the MUSICA project (Schneider
et al., 2012, 2016). This paired product is created using a post-processing step (1) to increase the sensitivity of the $\delta D$ data in
particular for dry conditions and (2) to harmonize the vertical sensitivities of the retrieved $H_2O$ and $\delta D$ products, such that both
profiles are representative for the overall same altitude regions, i.e. $H_2O$ and $\delta D$ have matching averaging kernels (Wiegele
et al., 2014; Schneider et al., 2016; Barthlott et al., 2017; Diekmann et al., 2021b). The MUSICA IASI $\{H_2O, \delta D\}$ pair product
is sensitive to variations of $H_2O$ and $\delta D$ in the free troposphere predominantly at around 4.2 km, with contributions from layers
up to 2–6 km (800–400 hPa). Diekmann et al. (2021b) have reported typical errors of up to $\sim 12\%$ for $H_2O$ and 30‰ for $\delta D$
at 4.2 km. The constellation of at least two simultaneously operating Metop satellites allows to achieve a twice daily global
coverage, with equator crossings at around 09:30 and 21:30 local time. Here, we consider only quality-checked $\{H_2O, \delta D\}$
pair data according to the recommended filter conditions for observations with high sensitivity, as defined in Diekmann et al.
(2021b).
The second product is the corresponding dataset from the Aqua/AIRS sensor (Worden et al., 2019). Similar to IASI, it captures
the thermal infrared part of the Earth's outgoing radiation, thus, analogously its $\{H_2O, \delta D\}$ product is also most sensitive to
water isotope abundances in the free troposphere. For the scope of this study $\{H_2O, \delta D\}$ data with a sensitivity peak between
135 825 and 421 hPa are selected. Errors reported for $\delta D$ are in a similar range as for IASI ($\sim 25 - 30\‰$). Conceptually, the overall
AIRS retrieval processor shares similarities with the MUSICA IASI retrieval, with however the difference that it deploys no
additional post-processing to achieve harmonization of the averaging kernels of the $H_2O$ and $\delta D$ retrieval states. In particular
for dry conditions, the retrieval results were found to be much more sensitive to variations in $H_2O$ than in $\delta D$, with the effect
that the averaging kernels of $H_2O$ and $\delta D$ showed significant discrepancies in their vertical structures. This might have the
140 consequence that the retrieved $H_2O$ and $\delta D$ results evaluated at the same retrieval grid level may not represent the same air
masses, so that their direct comparison would create misleading results. A more detailed discussion on these difficulties is
given in Schneider et al. (2016).

A third product of tropospheric water isotope data is retrieved from the short-wave infrared sensor Sentinel-5P/TROPOMI and recently published in the scope of the ESA S5P+Innovation Water Vapour Isotopologues (H2O-ISO) project by Trent et al. (2021). In contrast to IASI and AIRS, which capture infrared radiation emitted by the Earth's surface and atmosphere, TROPOMI measures solar radiation back-scattered at the Earth's surface. By making use of the TROPOMI observations only over land, where the back-scattering features of the short-wave infrared radiation are significantly more effective than over oceans, Trent et al. (2021) performed total-column-averaged dry-air mole fraction retrievals of $H_2O$ and HDO ($\delta D$ is then a posteriori calculated using Eqn.1). The uncertainty of the TROPOMI $\delta D$ product is estimated to achieve values of up to $\sim$ 30%, thereby being comparable to IASI and AIRS. The total-column-averaged $\delta D$ represents a weighted vertical $\delta D$ average, with the weighting performed according to the vertical $H_2O$ distribution, i.e. reported $\delta D$ values are determined by the most humid atmospheric layers. Consequently, we expect the column-averaged {$H_2O$, $\delta D$} products from TROPOMI to overall reflect the isotopic composition of the boundary layer or, in case of the WAM, the monsoon layer (i.e. the near-surface humid layer during the monsoon, which is typically deeper than the actual boundary layer). Further, we apply the quality filtering and bias correction for $H_2O$ and $\delta D$ as described and suggested in Trent et al. (2021).

As our study focuses on the information content of the retrieved water isotope observations and their implications on process understanding during the WAM, the reader interested in more technical information of the sensors and data products is encouraged to review the corresponding publications and documentations.

### 2.1.2 Cross-comparison of the different $H_2O$ and $\delta D$ datasets

For the scope of this study, we make use of the {$H_2O$, $\delta D$} dataset from IASI for its currently available period of 2015–2020, thereby resulting in a data overlap of five year with AIRS (available until 2019) and of almost three years with TROPOMI (available from April 2018 until December 2020). Figure1 shows the mean data availability of the considered data products over West Africa throughout the year, with focusing on data from February, May, August and November (the choice of these months is discussed in Section 5). Overall, the {$H_2O$, $\delta D$} pair dataset from IASI has a robust good coverage with up to $10^3$ observations per $1° \times 1°$ grid box per month. Observation counts are highest in the drier months February and November and lowest during rainy August, in particular over the precipitation maximum along the western coast and in Nigeria. AIRS has a data coverage comparably constant over time, however, with a significantly lower amount of available observations per grid point. In contrast to IASI and AIRS, TROPOMI data are available only over land, which is why we will omit the Guinea Coast domain for this sensor throughout this study. Further, a strong seasonal cycle in the data availability of TROPOMI becomes apparent, ranging from high to low data volumes from winter to summer. The fact that the reduced coverage correlates with the monsoon activity peak suggests that this is result of the cloud treatment in the TROPOMI retrieval processing.

As the isotopic datasets of both IASI and AIRS are vertically resolved with main sensitivity in the free troposphere, we can directly compare their daily $H_2O$ and $\delta D$ distributions for the Sahel and the Guinea Coast after interpolating the data to a common reference grid with $2.5°$ degrees grid size (see Fig. 2 and 3). Overall, the two satellites show great agreement for both $H_2O$ and $\delta D$, with, however, slight deviations for certain regimes. For instance, we observe that IASI depicts a pull to higher $H_2O$ ranges for moist conditions compared to AIRS. Further, we observe that for low $\delta D$ IASI appears more depleted than

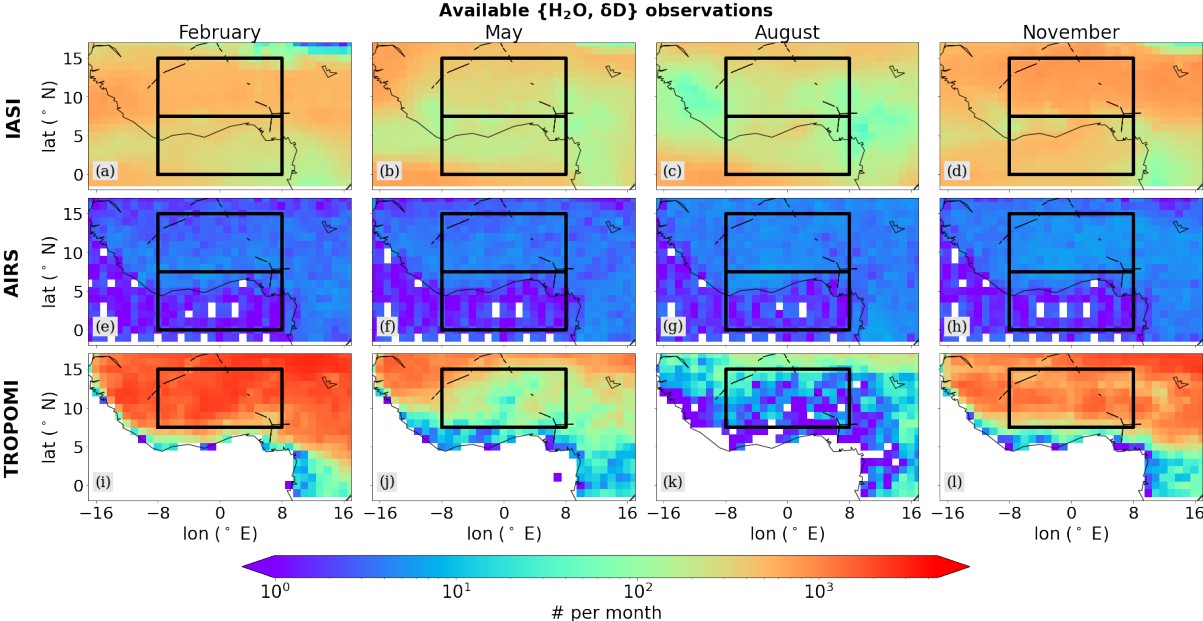

**Figure 1.** Horizontal distributions of the monthly averaged (from left to right February, May, August, November) coverage of available isotope retrievals from IASI, AIRS and TROPOMI evaluated on a $1° \times 1°$ grid, after filtering for clouds and data quality. The black frames indicate the study regions over the Sahel (upper box) and the Guinea Coast (lower box). As the TROPOMI dataset only consists of observations over land, the Guinea Coast region is omitted when studying the TROPOMI observations.

AIRS, which becomes particularly apparent over the Sahel during February and November. Such differences between the two datasets are expectable due to differences in the respective instrument and retrieval characteristics. For instance, IASI has a higher spectral resolution (0.5 cm−1) compared to AIRS (1 cm−1) and, hence, is more sensitive to atmospheric changes of $H_2O$ and $\delta D$. Further, the MUSICA IASI {$H_2O$, $\delta D$} pair retrieval includes various optimizations to harmonize and increase the sensitivities of $H_2O$ and $\delta D$, e.g. allowing to better capture changes for dry conditions (see Section 2.1.1).

In Fig. 4, we focus on the comparison of averaged absolute values per month and satellite product. We see that $H_2O$ from IASI appears consistently higher compared to AIRS, with an averaged difference of around $1000 - 1500$ ppmv. For $\delta D$, a positive bias by around $15 - 25$‰ becomes visible in the AIRS data compared to IASI.

An interpretation of the comparison of IASI and AIRS with TROPOMI data is difficult, because the data represent different altitude ranges (i.e. total columns versus free troposphere) and $H_2O$ has a strong gradient from the boundary layer to the troposphere .In terms of $\delta D$, TROPOMI appears even more enriched on the large scale than IASI and AIRS ($\delta D$ increase up to $70$‰). This is expected as TROPOMI is a total column product and thus dominated by the lower troposphere, which is typically more enriched due to contributions from e.g. evapo-transpiration (Worden et al., 2021). In general, this direct comparison of the absolute data range of $H_2O$ and $\delta D$ between TROPOMI (total column products), IASI and AIRS (mid-tropospheric

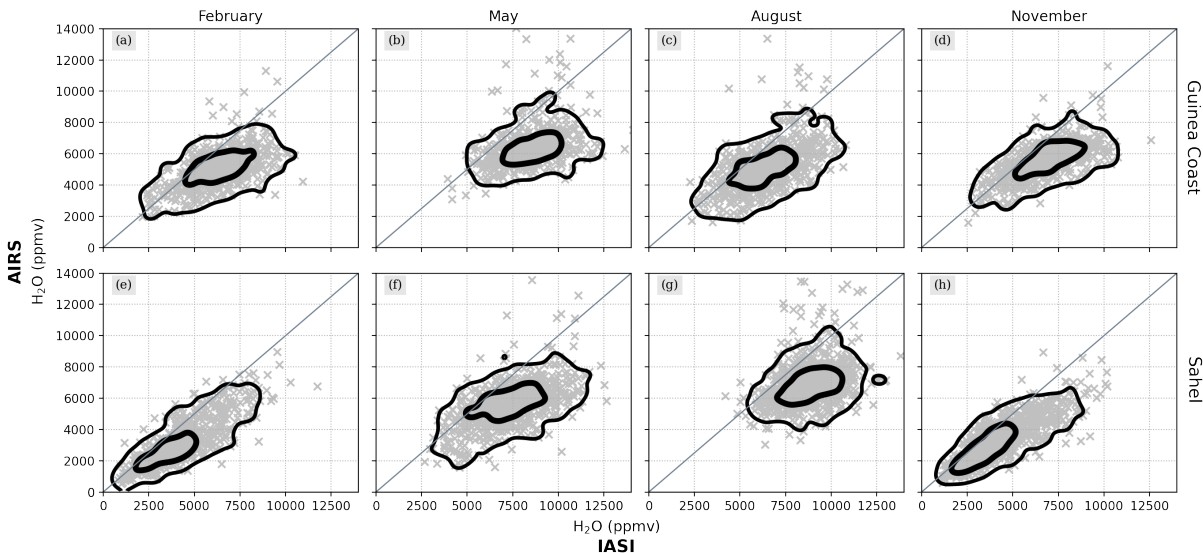

**Figure 2.** Comparison of daily $H_2O$ data from IASI against AIRS over the Guinea Coast and the Sahel for February, May, August and November from the years 2015 – 2019. The satellite data have been re-gridded onto a regular $2.5° \times 2.5°$ grid by averaging all samples within a single grid box. The gridded data are then cross-compared for IASI and AIRS, shown as grey scatter and described by the two-dimensional histogram contours.

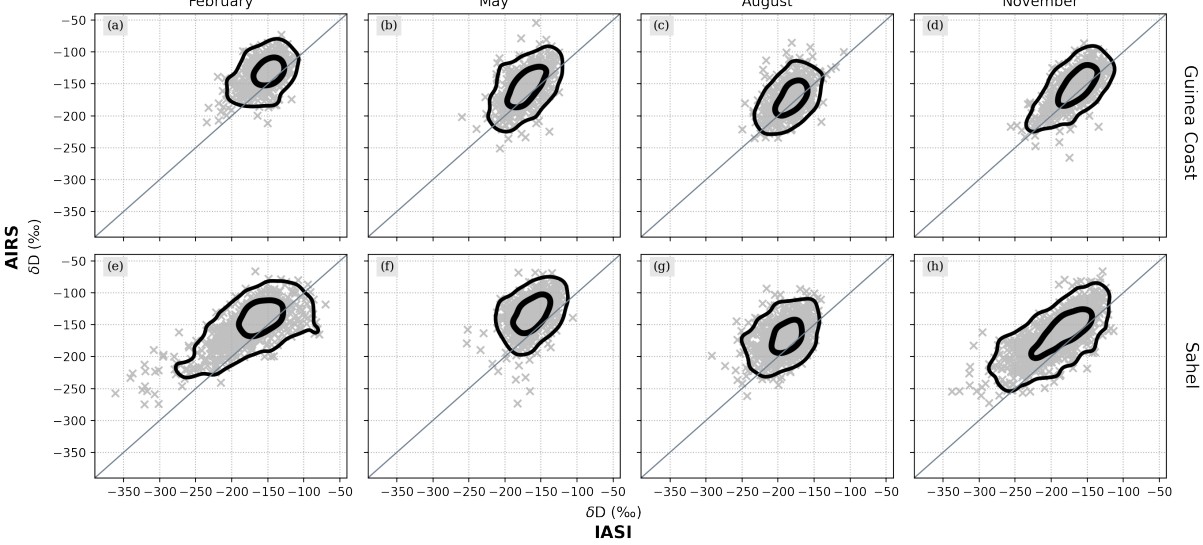

**Figure 3.** Same as Fig. 2, but for corresponding $\delta D$ data.

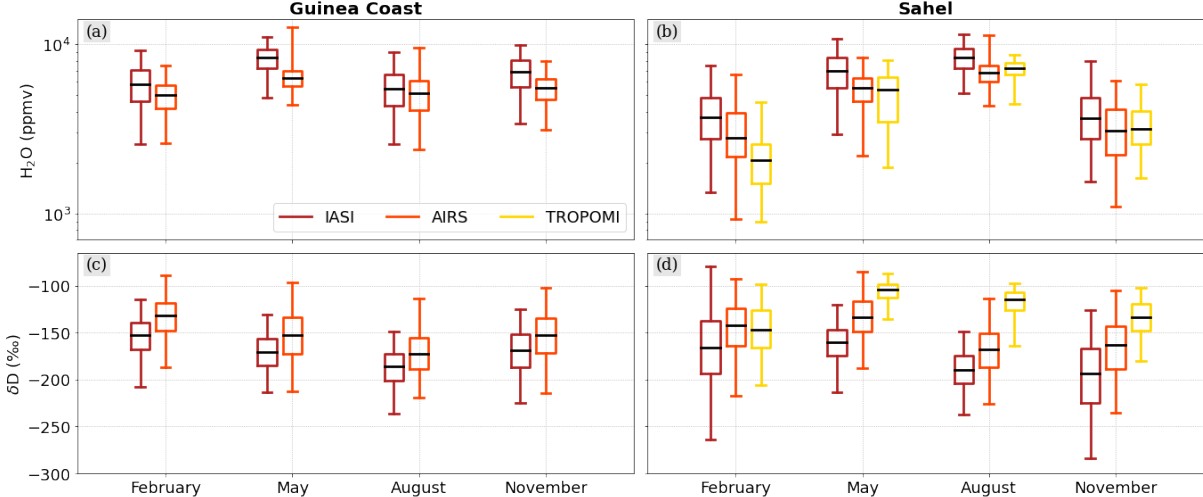

**Figure 4.** Boxplots evaluating the distributions of $H_2O$ (using the natural logarithm) and $\delta D$ from IASI, AIRS and TROPOMI for the Guinea Coast (a, c) and the Sahel (b and d), averaged for February, May, August and November from the available periods, respectively (see Table 1). The boxes extend from the lower to the upper quartile with the middle line depicting the median.

products) needs to be treated with caution due to the structurally differing vertical sensitivities of the different products.

15 Schneider et al. (2016), Schneider et al. (2016) As a next step of cross-comparing the different satellite datasets, we now focus on the distribution shapes of $H_2O$ and $\delta D$ from a qualitative point of view, as shown by the respective probability density

functions (PDF) in Fig. 5. We observe that in particular IASI and AIRS show great agreement for both $H_2O$ and $\delta D$, with the largest discrepancy again for the Sahelian February data. As has been discussed above, IASI reaches significantly lower $\delta D$ values for dry regimes, which has the effect that the corresponding PDF stretches and flattens compared to AIRS. Further, the TROPOMI PDFs depict overall agreement with IASI and AIRS for $H_2O$, however, reveal discrepancies for $\delta D$, e.g. in terms of relative PDF positions between February to May and between February to August. This results from the different character-

istics of the TROPOMI data being total column products compared to the mid-tropospheric IASI and AIRS products. To better understand such differences with respect to the underlying control processes (see Section 2.3), this study will shed light on the paired $\{H_2O, \delta D\}$ distributions of the three instruments.

## 2.2   Precipitation products from remote sensing

In order to obtain information about precipitation over West Africa, we make use of the IMERG precipitation dataset, which is a fused product from precipitation estimates retrieved during the TRMM (Tropical Rainfall Measuring Mission) and GPM (Global Precipitation Mission) satellite missions (Huffman et al., 2014). The retrieved precipitation estimates are calibrated with actual rain gauge data on a monthly basis and the final IMERG precipitation product is available on a half-hourly, daily and monthly basis. The daily precipitation product is derived from averaging half-hourly precipitation rates at the considered

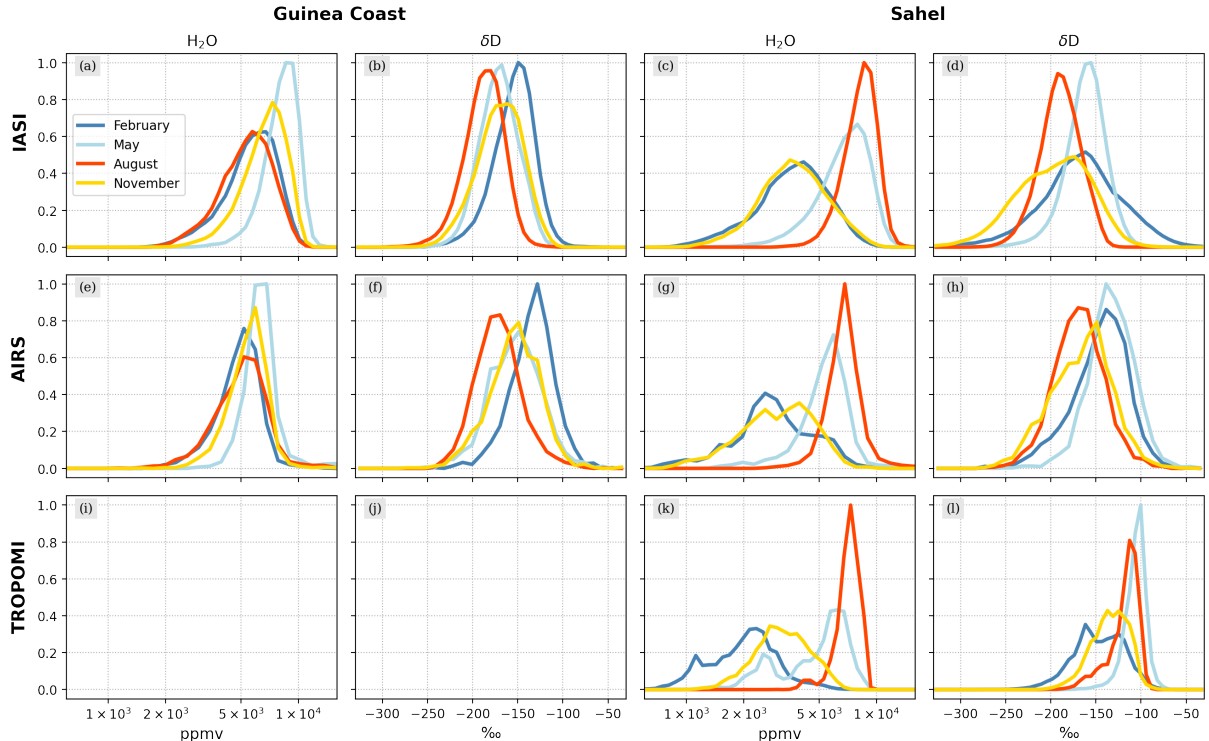

**Figure 5.** Probability density distributions for the $H_2O$ and $\delta D$ data shown in Fig. 15.

0.1° ×0.1° grid and afterwards multiplying the mean precipitation rates (in mm per hour) by 24 to obtain daily rates (i.e. mm per day) (Huffman et al., 2019). Using long-term series of several rain gauges, Maranan et al. (2020) have performed a systematic and process-based validation of the IMERG precipitation product for the West African zone and, hence, have demonstrated that IMERG is capable of reproducing the well-known evolution of West African precipitation over the full year.

The high resolution in space and time of the remotely sensed {$H_2O$, $\delta D$} pairs from the sensors described in Sect. 2.1.1 allows to examine the impact of individual convective systems on tropospheric water isotope signals. As the considered IASI, AIRS and TROPOMI datasets are limited to cloud-free scenes, we cannot measure directly the impact of microphysical rain processes occurring within convective cells. Therefore, we utilize the IMERG precipitation dataset to derive a simple clustering method, which allows to assess, whether the observed cloud-free air masses have been affected in their recent history by convective precipitation. In this context, we aim to categorize the available satellite observations as follows: (1) If within a chosen grid box and at a given calendar day the daily precipitation summed over the previous three days remains below a chosen threshold (here, 0.2 mm per day), then all observations inside this grid box and at the target date are classified as *non-rain* observations. (2) If one day prior to the given calendar day a chosen grid box experiences an averaged precipitation rate higher than a chosen threshold (here, 3 mm per day), then all observations within this grid box and at the target date are classified as *post-rain*. For this purpose, we consider the daily IMERG precipitation data to cluster the daily {$H_2O$, $\delta D$} pairs from all three sensors at

a target grid size of $1° \times 1°$. Despite the rather simple nature of this approach (e.g. neglecting effects such as the advection of air masses from adjacent grid boxes, which could be addressed by Lagrangian backward trajectories), the current choice of method and thresholds proves successful to identify the targeted precipitation states, i.e. persistent dry conditions against convective systems with marked precipitation (see Sect. 3).

By utilizing the results of the precipitation clustering method, we derive a decomposition method that assesses to which extent the non-rain and post-rain clusters may prove useful to explain year-to-year variations in $\delta$D as observed in the satellite datasets. For this purpose, we focus on $\Delta\delta D$ as the overall year-to-year anomaly in $\delta$D and decompose it by estimating the contribution of the following control factors per year $i$:

$$\Delta\delta D_i = \Delta\delta D_{\mathrm{r},i} + \Delta\delta D_{\mathrm{non\text{-}rain},i} + \Delta\delta D_{\mathrm{post\text{-}rain},i} + c \qquad (2)$$

with

$$\Delta\delta D_{\mathrm{r},i} = \left( f_{\mathrm{post\text{-}rain},i} - \overline{f}_{\mathrm{post\text{-}rain}} \right) \cdot \left( \overline{\delta D}_{\mathrm{post\text{-}rain}} - \overline{\delta D}_{\mathrm{non\text{-}rain}} \right)$$

$$\Delta\delta D_{\mathrm{non\text{-}rain},i} = \overline{f}_{\mathrm{non\text{-}rain}} \cdot \left( \delta D_{\mathrm{non\text{-}rain},i} - \overline{\delta D}_{\mathrm{non\text{-}rain}} \right) \qquad (3)$$

$$\Delta\delta D_{\mathrm{post\text{-}rain},i} = \overline{f}_{\mathrm{post\text{-}rain}} \cdot \left( \delta D_{\mathrm{post\text{-}rain},i} - \overline{\delta D}_{\mathrm{post\text{-}rain}} \right)$$

Here, $\Delta\delta D_{\mathrm{r}}$ represents the impact of year-to-year anomalies in the fraction of post-rain events, $f_{\mathrm{post\text{-}rain}}$. Analogously, $\Delta\delta D_{\mathrm{non\text{-}rain}}$ refers to the contribution to $\Delta\delta D$ due to anomalies in $\delta D_{\mathrm{non\text{-}rain}}$ and $\Delta\delta D_{\mathrm{post\text{-}rain}}$ due to anomalies in $\delta D_{\mathrm{post\text{-}rain}}$. $f_{\mathrm{non\text{-}rain}}$ is the fraction of non-rain events. The variables marked with a bar refer to the averaged value over all considered years, and the variables with the subscript $i$ denote the respective values per year. $c$ refers to the residual anomalies in $\delta$D that cannot be addressed by the considered clustering method (e.g. the non-rain and post-rain clusters cover only the edge cases of precipitation rates and not the range in between) and, hence, is not part of this study. Consequentially, the described decomposition method will not provide a complete view on year-to-year anomalies in $\delta$D, but will focus on the separate contributions of the described control factors.

## 2.3 Process curves in the {H$_2$O, $\delta$D} phase space

Further, we make use of idealized process curves derived by Noone et al. (2011) that help interpret observed {H$_2$O, $\delta$D} pair distributions with respect to the underlying dynamical and microphysical processes. In particular, Diekmann et al. (2021a) have applied these process curves to describe characteristic {H$_2$O, $\delta$D} pair variations with respect to processes representative of the WAM (see Fig. 6): (1) The hyperbolic curves indicate characteristic {H$_2$O, $\delta$D} signals due to mixing of dry/depleted and humid/enriched air masses. (2) The Rayleigh lines mark the depletion in $\delta$D according to condensation and subsequent moisture removal by rainout. If evaporation and equilibration occur on top of a Rayleigh process, then enhanced depletion in $\delta$D takes place and leads to the so-called Super-Rayleigh regime below the Rayleigh lines (3). This is found to be particularly pronounced over the Sahel during the WAM season, when well-organized convective systems such as squall lines lead to intense convective rainfall.

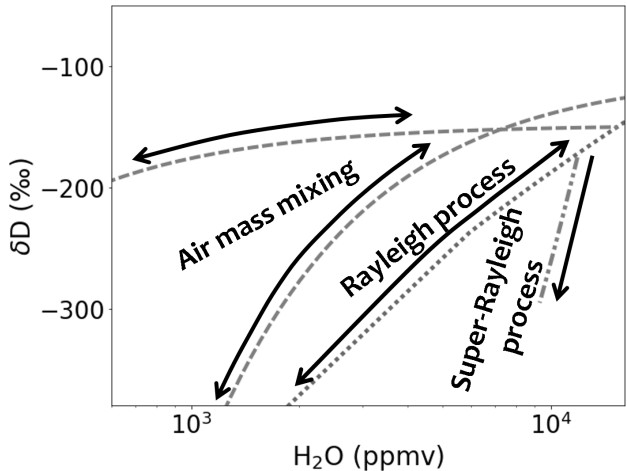

**Figure 6.** Theoretical process curves describing effects of idealized dynamical and microphysical processes in the $\{H_2O, \delta D\}$ phase space. The curves are defined as described in Diekmann et al. (2021a): The hyperbolic mixing curves are representative of dry and moist air mass mixing processes over West Africa, considering the mixing members $x_0 = [50 \text{ ppmv}, -700 \text{ ‰}]$ and $x_1 = [15300 \text{ ppmv}, -120 \text{ ‰}]$ for the dry mixing curve (upper curve) and $x_0 = [100 \text{ ppmv}, -450 \text{ ‰}]$ and $x_1 = [22000 \text{ ppmv}, -120 \text{ ‰}]$ for the moist mixing curve (lower curve). The Rayleigh curve is resulting from a Rayleigh process with initial conditions $\delta D_0 = -80 \text{ ‰}$, RH = 90 % and $T_0 = 30° \text{ C}$. The Super-Rayleigh curve has been defined in order to represent signals due to rain evaporation and equilibration as found along backward air trajectories by Diekmann et al. (2021a).

Throughout this study, the idealized curves identified in Diekmann et al. (2021a) will provide a process-based interpretation context for the $\{H_2O, \delta D\}$ pair variability as observed by the remote sensing sensors.

## 3 Convective-scale variability of $\{H_2O, \delta D\}$ pair signals

In a first step, we aim to address the variability of the spaceborne $\{H_2O, \delta D\}$ pair observations with respect to microphysical processes that are associated with convection on a daily event basis. It applies the clustering method from Sect. 2.2 to the $\{H_2O, \delta D\}$ data from IASI, AIRS and TROPOMI to identify those observations that were impacted by considerable monsoon precipitation and to contrast them to largely dry instances. In the following, this is done for a case study focusing on the season 2016 and additionally for a statistical multi-year analysis.

### 3.1 Rain analysis of West African Monsoon 2016

We demonstrate the basic concept of the clustering approach into non-rain and post-rain events and evaluate it on a case-study basis using the IASI data during the West African Monsoon in 2016. This specific monsoon season was the key subject of the multi-platform DACCIWA campaign (Knippertz et al., 2017), where data were collected over the period June – July 2016 in

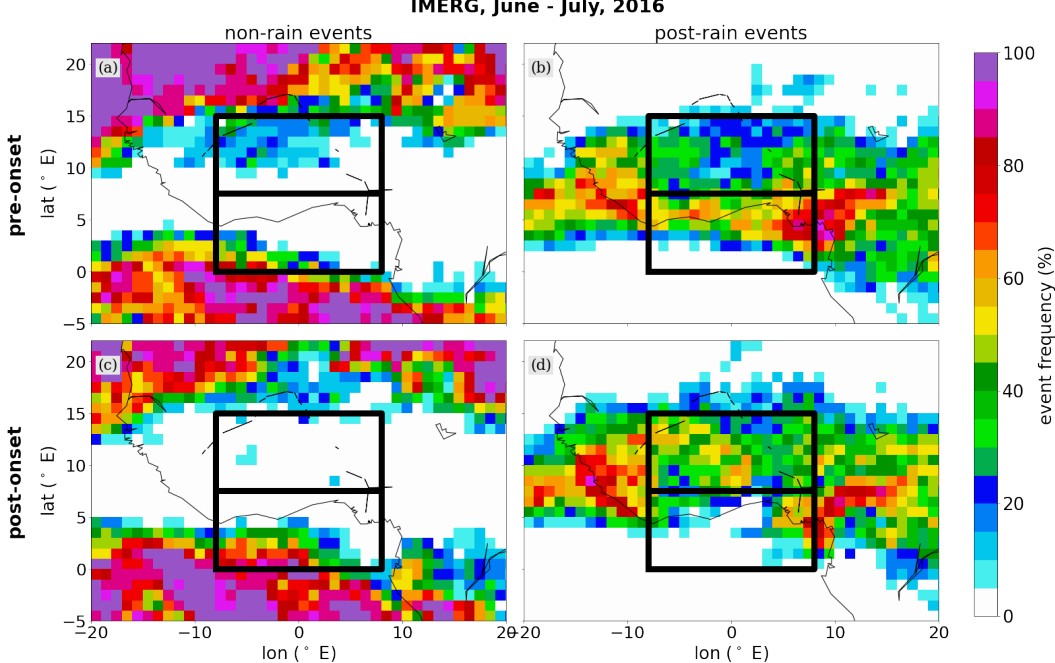

**Figure 7.** Fraction of days marked as (a, c) non-rain and (b, d) post-rain events for the pre- and post-onset phases of the WAM between June – July 2016 (with the monsoon onset occurred on 22 June 2016), derived by considering the daily averaged precipitation from the half-hourly IMERG precipitation dataset and evaluated on a $1° \times 1°$ grid. The black-framed boxes depict the chosen target domains over the Guinea Coast and the Sahel.

order to characterize the monsoon system from synoptic, microphysical, dynamical and aerosol perspectives. In this framework, the onset of the monsoon activity was defined according to the northwards shift of precipitation from the Guinea Coast to the Sahel and was set to 22 June 2016. Based on this, we separate the period into the pre-onset (01 – 21 June 2016) and the
post-onset phase (22 June – 31 July 2016), with the aim to further underline the impact of the WAM activity to the $\{H_2O, \delta D\}$ pairs.

Figure 7 provides a horizontal overview about the relative fraction of days that are identified as non-rain or post-rain events during the pre- and post-onset phases, for each considered $1° \times 1°$ grid box over West Africa. Overall, this clustering appears to successfully capture the areas associated with strong precipitation. It well reflects the onset criterion for the WAM activity,
i.e. before the onset the predominance of precipitation is located over the Guinea Coast with occasional convective events over the Sahel, while after the onset the precipitation maximum moves to the Sahel. The post-onset phase leaves the Guinea Coast mostly rain-free with regional exceptions for its easternmost areas, where around the coasts of Nigeria and Cameroon, a localized and persistent precipitation peak prevails (e.g. discussed in Nlend et al., 2020).

Figure 8 shows the IASI $\{H_2O, \delta D\}$ distributions sorted into non-rain and post-rain clusters for the phases before and after the
monsoon onset. During the pre-onset phase, both regions feature a considerable number of events for both non- and post-rain

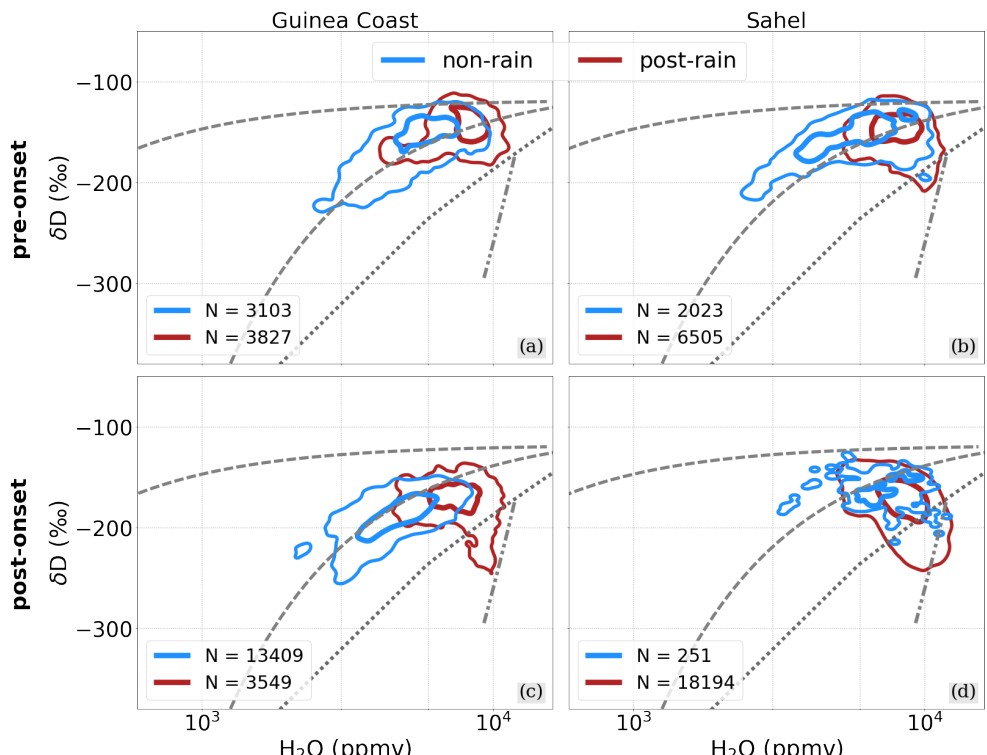

**Figure 8.** IASI $\{H_2O, \delta D\}$ pair data over the Guinea Coast and the Sahel for the pre- and post-onset WAM phases between June – July 2016 (monsoon onset on 22 June 2016), clustered into non-rain and post-rain events based on daily averaged precipitation from IMERG. The $\{H_2O, \delta D\}$ pair data are summarized by normalized two-dimensional histogram contours, with the contours indicating the main 95 % and 50 % of the respective data points and with the corresponding total data numbers given in the legend (calculated according to Eckstein et al. (2018)).

classes. It becomes apparent that the non-rain events from both Guinea Coast and Sahel consistently evolve along the mixing line, which is in alignment with our understanding that no microphysical interactions have occurred and that the respective air masses are instead dominated by mixing. As discussed in the trajectory study from Diekmann et al. (2021a), air masses from different origins meet over West Africa, mainly the moist southerly monsoon flow from the Gulf of Guinea, the dry Saharan northerly winds as well as dry intrusions from the upper troposphere.

As a response to the formation of strong precipitation systems over the Guinea Coast, we observe that the corresponding $\{H_2O, \delta D\}$ pairs mainly move upwards the mixing line to higher $H_2O$ and $\delta D$ regimes with no observable Rayleigh and Super-Rayleigh features. This can be explained by the fact that rain evaporation is reduced for more intense precipitation events (Dansgaard, 1964). As large rain drops fall faster, there is less time for evaporation, leading to less depletion of the ambient vapour (Stewart, 1975). Additionally, the rainfall over the Guinea Coast appears to be sufficiently large-scale and per-

sistent such that the moist updrafts lead to a significant moistening of the free troposphere, which in turn leads to high relative humidity and thus further reduces the rain evaporation efficiency (Risi et al., 2008a; Lee and Fung, 2008; Yoshimura et al., 2010).

In contrast, over the Sahel a noticeable pull towards the rain-evaporation driven Super-Rayleigh regime stands out, forming a clearly anti-correlated relation between $H_2O$ and $\delta D$. The precipitation events appear to be less intense than over the Guinea Coast (see Fig. 7), but over the Sahel strong intrusions from the Sahara and the upper troposphere transport dry and unsaturated air masses into the troposphere, hence increasing the rain evaporation efficiency. This effect intensifies during the post-onset phase, when precipitation is mainly driven by westward propagating convective events, so-called squall lines that induce strong convective precipitation in the leading edge of the system and moderate precipitation below the trailing stratiform cloud shield (e.g. Risi et al., 2008b, 2010b; Lafore et al., 2017; Diekmann et al., 2021a). The latter is characterized by smaller rain drops that, as discussed above, have in general a higher rain evaporation efficiency. Additionally, such squall line systems are driven by a rear-to-front inflow (usually connected to the African Easterly Jet), which feeds dry and unsaturated air to mid- and lower levels below the stratiform rain region and thus further increases the efficiency of rain evaporation. In this way, squall lines exert a strong fractionating effect on the West African troposphere, which is reflected in clear Super-Rayleigh features in the post-rain $\{H_2O, \delta D\}$ pairs over the Sahel and slightly over the Guinea Coast after the monsoon has set in. This is in agreement with Diekmann et al. (2021a), who attributed this Super-Rayleigh regime to microphysical rain processes within the melting zone of a convective system, namely the evaporation of falling rain drops in unsaturated areas as well as the equilibration between water vapour and relatively depleted rain drops formed from melting snow for saturated conditions (see also Risi et al., 2021).

## 3.2 Multi-year analysis

After characterizing the predominant control processes of the IASI $\{H_2O, \delta D\}$ pairs during June – July with respect to the monsoon onset during 2016, the objective of this section is to provide a statistically more robust view on the observed convection-related $\{H_2O, \delta D\}$ features. For this purpose, we now aim for a multi-annual view on $\{H_2O, \delta D\}$ signals from the June – July average clustered for the non-rain and post-rain events and evaluated for all available data for all three sensors, IASI, AIRS and TROPOMI (see Table 1). The $\{H_2O, \delta D\}$ pairs collected and clustered for the Guinea Coast and the Sahel are shown in Fig. 9, except for TROPOMI over the Guinea Coast (see Fig. 1). The corresponding boxplots for $H_2O$ and $\delta D$ of the non- and post-rain clusters (shown in Fig. 10) serve to add a more quantitative perspective on the differences between the signals from the two clusters.

It becomes apparent that the characteristic $\{H_2O, \delta D\}$ signals of the two rain clusters identified for IASI during 2016 (Fig.8) persist as robust feature over the multi-annual perspective for both IASI and AIRS: While the Guinea Coast experiences an overall increase in both $H_2O$ (around 5 %) and $\delta D$ (15 – 25 ‰) along the mixing curves, when comparing the non- to post-rain events, the Sahel develops the previously discussed anti-correlated $\{H_2O, \delta D\}$ features towards the Super-Rayleigh regime as robust response to convective monsoon precipitation (1.5 – 3 % increase in $H_2O$, 10 – 25 ‰ decrease in $\delta D$).

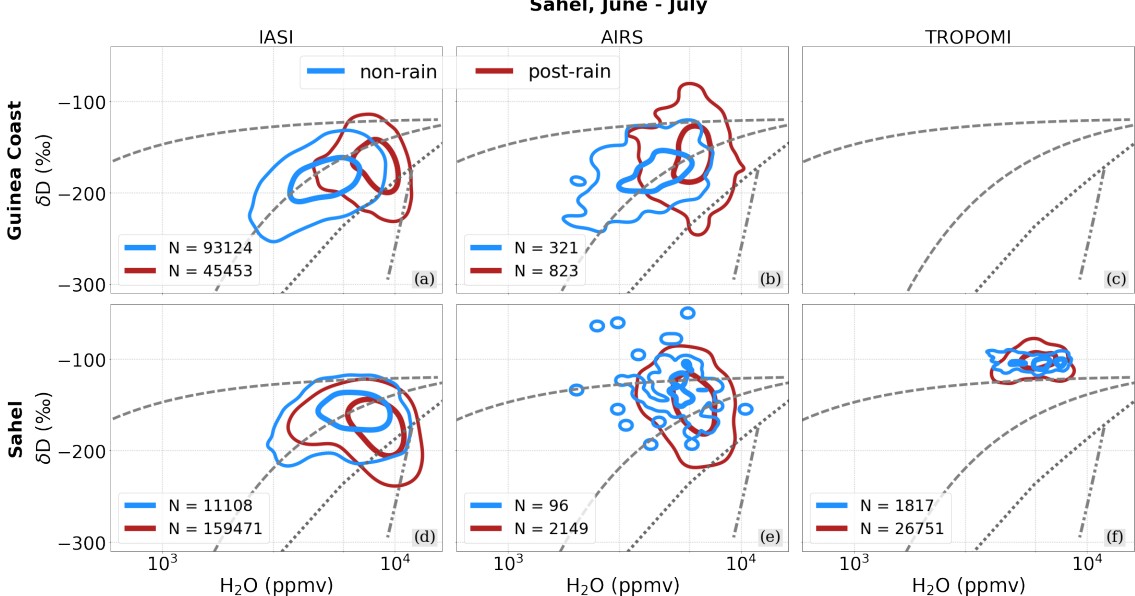

**Figure 9.** {$H_2O$, $\delta D$} pair data from IASI and AIRS over the Guinea Coast (a and b) and from IASI, AIRS and TROPOMI over the Sahel (d, e and f) as average for June – July for the respectively available years (see Table 1), clustered into non-rain and post-rain events based on daily averaged precipitation from IMERG and with the contours depicting the distribution of the main 95 % and 50 % scatter points.

In contrast to the vertically resolved data products from IASI and AIRS, TROPOMI does not exhibit such a clear distinction between the non- and post-rain events over the Sahel, but rather experiences a broadening of the {$H_2O$, $\delta D$} distribution along the $\delta D$ dimension. 10), 2.1.1) (Noone, 2012; Worden et al., 2021) (Risi et al., 2008a; Diekmann et al., 2021a), (Diekmann et al., 2021a). . 9). Accordingly, we find that the boxplots of the non-rain and post-rain classes shown in Fig 10, reveal a high level of agreement in terms of their mean values. This is contrary to various studies that have documented significant drying and depletion of near-surface air after the passage of a convective squall line event as result of dry rear-to-front downdrafts (e.g. Tremoy et al., 2012, 2014; Lafore et al., 2017; Torri, 2022; Risi et al., 2023). The resulting warm and dry air near the surface enhances the evapo-transpiration of surface water, which is associated with enrichment in $\delta D$ (Noone, 2012; Worden et al., 2021) and which thus may accelerate the recovery of water vapor after the convective event. The advection of moist air masses due to the monsoon flow from the Tropical Atlantic into the Sahel further foster the recovery of the post-storm near-surface air (Lafore et al., 2017). Therefore, we assume that a rapidly recovering water vapor may hamper the capability of the clustering to capture actual post-rain conditions (e.g. Super-Rayleigh signals that are typical for atmosphere affected by convection), when applying to TROPOMI and defining the post-rain events by means of the precipitation from the previous day. Instead, a higher temporal frequency would be beneficial to create more direct links between individual convective systems and collocated TROPOMI data.

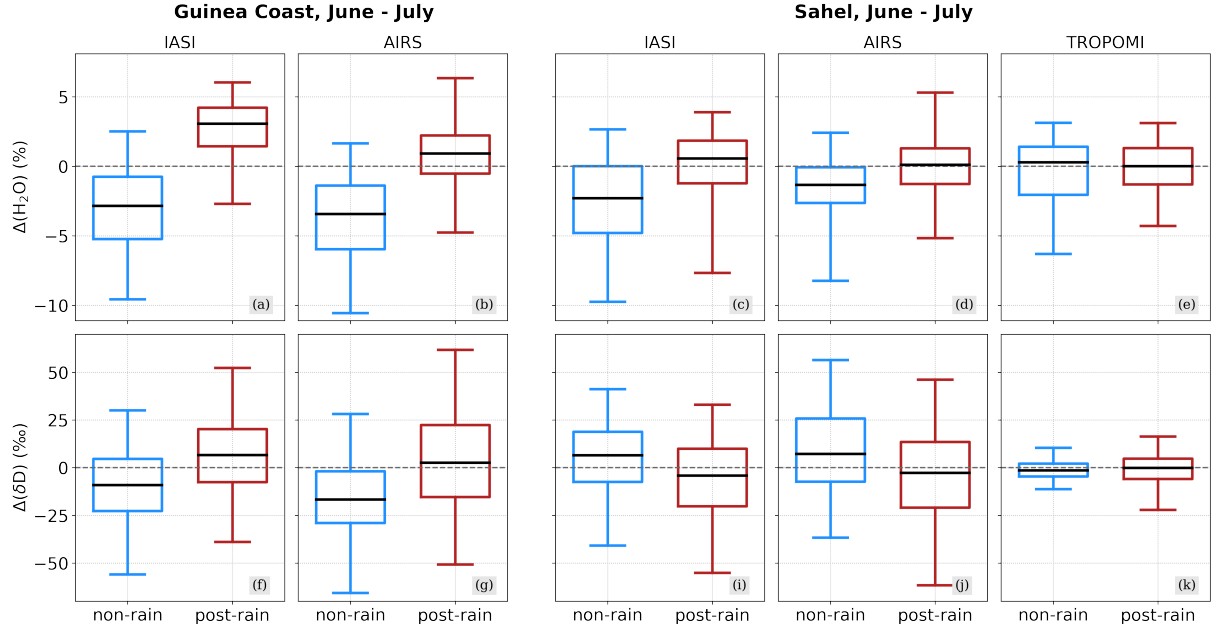

**Figure 10.** Boxplots evaluating the distributions of $H_2O$ and $\delta D$ for the non-rain and post-rain clusters during June – July for the respective available years from IASI (a, c, f, i), AIRS (b, d, g, j) and TROPOMI (e, k), see Table 1. The boxes extend from the lower to the upper quartile values with the middle line depicting the median. The whiskers mark the lower and upper 2.5 percentiles of the underlying distributions. The $H_2O$ (evaluated using the natural logarithm) and $\delta D$ values are given as difference to the corresponding instrument-specific average over the full available period (see Table 1).

### 3.3 Decomposition of multi-year $\delta D$ anomalies

From Fig. 10 we can estimate the parameters required for applying the decomposition method for year-to-year anomalies in $\delta D$, as described in Sect. 2.2. In particular, this includes the $\delta D$ values for the individual years and the multi-year average for the non-rain and post-rain clusters as well as the per-year and averaged fractions of non-rain and post-rain events. The $\delta D$
anomalies resulting from variations in rainfall fraction ($\Delta \delta D_r$) as well as from variations in $\delta D$ for non-rain and post-rain are shown per instrument and year in Fig. 11.

The results indicate that for the considered years the anomalies in $\delta D$ of the post-rain situations exhibit high year-to-year variability to the overall $\delta D$ anomaly, while respective anomalies in rainfall fractions as well in $\delta D_{non-rain}$ show only minor relative contribution to $\Delta \delta D$. Since the latter implies that rainfall fraction has been overall stable during the considered time
period, this lets us conclude that the strong variations in $\delta D_{post-rain}$ are resulting from convective events with varying intensity, in terms of e.g. precipitation rates and dry air intrusions. The differences in $\Delta \delta D_{post-rain}$ observed between IASI and AIRS may arise from the different sampling. With the spatial coverage of AIRS being much lower than IASI, it makes the statistical analysis of AIRS more sensitive to outliers, while for IASI the results become smoother. This would explain why for positive

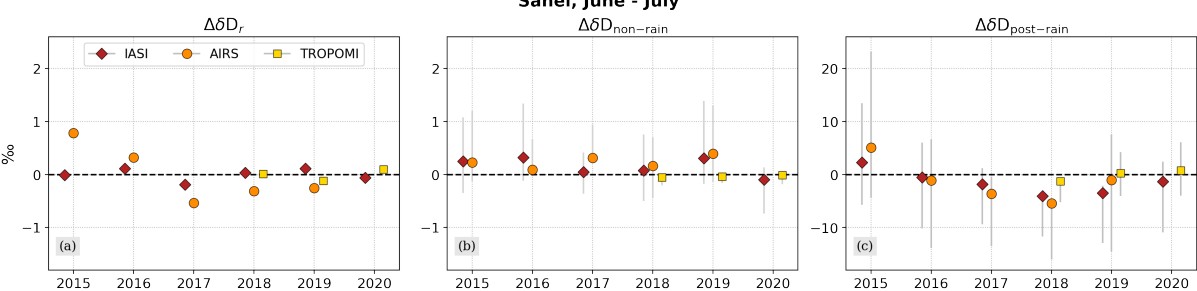

**Figure 11.** Results from the decomposition analysis to characterise the impact of the anomalies in rain fraction (a), non-rain events (b) and post-rain events (c) on the overall $\delta$D anomalies for IASI, AIRS and TROPOMI for the respectively available years. The scatters represent the mean values of the respective anomaly contributions and the grey bars denote the $1\sigma$ spread around the mean (only plotted for non-rain and post-rain). The ranges of the y-axes have been individually selected with respect to the data variability per sub-figure. A detailed description of the decomposition method is given in Sect. 2.2.

$\Delta\delta$D$_{\text{post-rain}}$ values, AIRS is higher than IASI and for negative values it generally appears lower.

The overall low contributions of $\delta$D$_{\text{non-rain}}$ to $\Delta\delta D$ in all three datasets is driven by the much lower fraction of non-rain events compared to post-rain (see Fig. 7).

### 3.3.1 Discussion of the clustering method

This section demonstrates that the rather simple and observational-based clustering approach allows to successfully identify

observations with and without substantial impact due to convective precipitation, and in addition to clearly distinguish them in the {H$_2$O, $\delta$D} phase space in alignment with the characteristic signals due to air mass mixing and microphysical rain-vapour interactions.

A main outcome of this analysis is that it reveals significant regional differences on how convective events impact the {H$_2$O, $\delta$D} pair variability, i.e. whether they have an enriching or depleting effect on mid-tropospheric $\delta$D. Thus, a promising per-

365 spective to strengthen the value of the clustering results would be to characterize post-rain clusters as function of precipitation rates, with the aim to address the role of convection type on the isotopic composition of tropospheric water vapour. Maranan et al. (2018) have found that convection over Southern West Africa may form in various types, ranging from highly organized Mesoscale Convective Systems to small-sized isolated systems. Further, Galewsky et al. (2023) have linked such differences in convective aggregation to features in $\delta$D from IASI. They found that highly aggregated convective systems are character-

370 ized by mostly moderate precipitation and less pronounced $\delta$D depletion in tropospheric water vapour, whereas unaggregated convection with strong precipitation is associated with more depleted $\delta$D signals. This is somewhat contradictory to the result from our cluster analysis as well as from other studies (Risi et al., 2008b; Tremoy et al., 2014; Diekmann et al., 2021a) that mid-tropospheric $\delta$D appears relatively more depleted after the passage of Sahelian squall lines, . While squall lines are

highly organized systems and, hence, represent aggregated convection, they are associated with deep convective precipitation in the leading edge followed by a trailing stratiform cloud shield with moderate precipitation. Thus, this demonstrates that more refined methods are required to achieve a more comprehensive view on convection and its impact on $\delta$D, for instance by combining methods addressing convective aggregation (such as from Galewsky et al. (2023)) and convective depth (e.g., by introducing further and specifically tailored rainfall selection criteria to our clustering method).

Further, we have put the focus of the clustering method to the monsoon onset phase, with the aim to characterize the impact due to convection as opposed to the impact due to air mass mixing. Therefore, another interesting question for future studies would be to investigate, whether the clustering method would provide the same results for the monsoon retreat phase or whether asymmetries would appear, e.g. due to different land-sea contrasts during the different seasons and seasonal mean state differences. For instance, Diaconescu et al. (2015) have shown that atmospheric models and reanalyses show significant differences in the performances of reproducing the monsoon onset and retreat. Due to the impact of convection, simulating the monsoon onset remains a key challenge for the models, while the modelled retreat phase is more in line with observational datasets. This points to differences in the main processes controlling the atmospheric state over the Sahel and, for this purpose, our clustering method would be valuable to evaluate to which extent dynamical and microphysical processes change from an isotopic perspective when compared to the onset.

Galewsky et al. (2023)

## 4 Inter-annual variability of convective {$H_2O$, $\delta$D} signals during the peak monsoon

In a second step, we investigate, to which extent the short-term convection-related {$H_2O$, $\delta$D} features identified on an event basis are observable on a monthly scale comparing different years. For this purpose, we focus on the isotopic signals in the Sahelian mid-troposphere during the peak summer monsoon in August. Figure 12 exhibits the median and data spread for $H_2O$ and $\delta$D from IASI and AIRS as anomalies to the average of all available August data (TROPOMI is omitted due to the limited time coverage) and for precipitation from IMERG.

The first half period, 2015 – 2017, is characterized by a stable median of $H_2O$ with both IASI and AIRS agreeing with their multi-annual averages. Differences between the two datasets arise with respect to the data spread, which, however, needs to be interpreted carefully, since IASI and AIRS have a considerably different amount of available observations (on average, differing by a factor of 50–200, see Fig. 1) and distributions with smaller samples are more sensitive to outliers. Further, in reasonable agreement between both sensors, $\delta$D is enriched between 2015 – 2017 compared to the full period. The annual averages exceed the full period average with up to 20‰ and the upper data spread limits reach up to 60 – 70‰ above average.

The second half of the full period, 2018 – 2020, experiences a marked increase in the spread of the precipitation. The year 2018 has the highest precipitation median value of the six study years, although $H_2O$ does not depict a significant deviation from the average. In contrast, the IASI $\delta$D shows a consistent drop of down to 25‰ compared to 2017 with respect to the median value and data spread. While this decrease in $\delta$D is not observed for AIRS during 2018, it becomes even more apparent in 2019, where both sensors indicate a larger depletion in $\delta$D compared to 2017. The $\delta$D averages reach up to 10‰ below

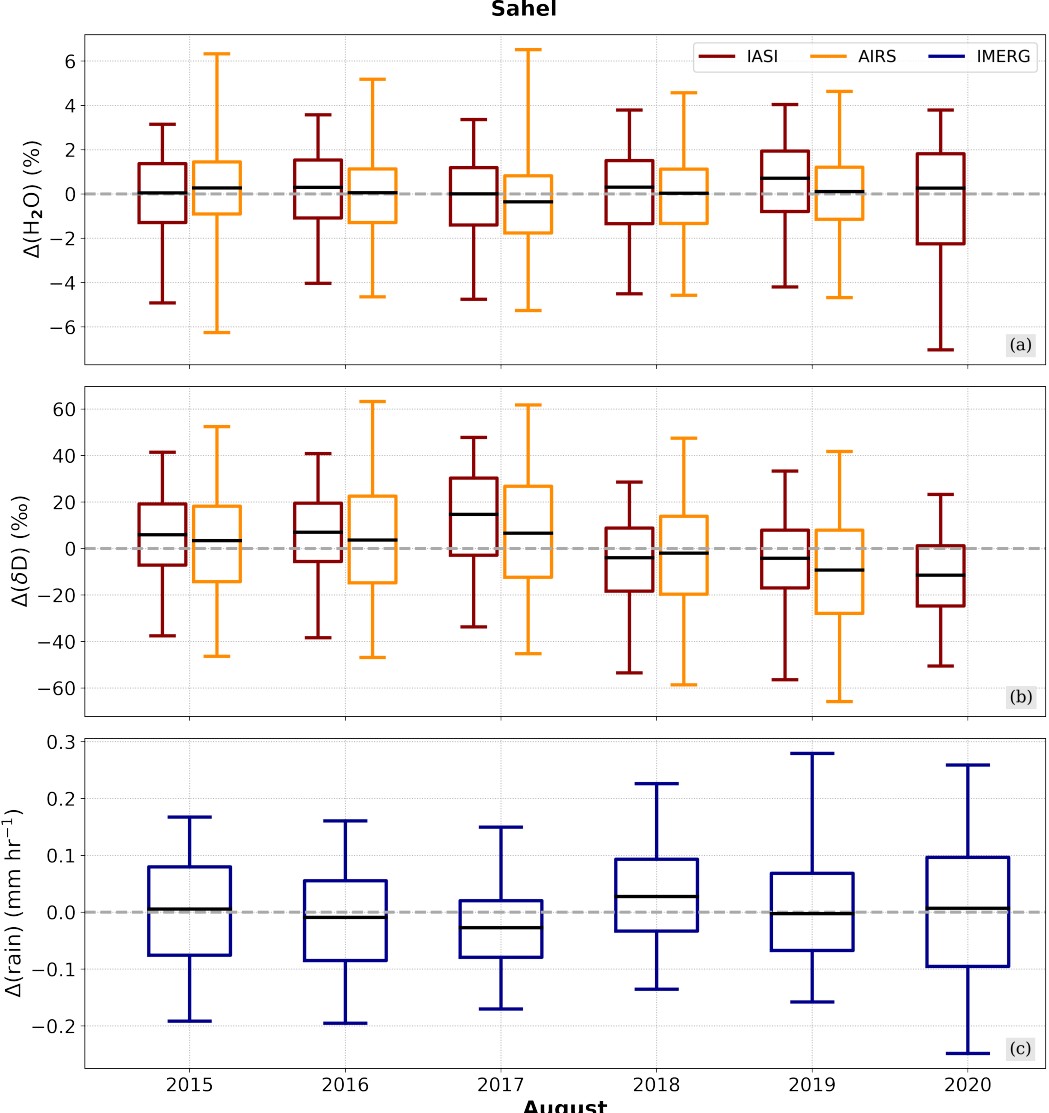

**Figure 12.** Boxplots evaluating the August distributions of $H_2O$ and $\delta D$ from IASI and AIRS (red and orange boxes, shown in a, b) and precipitation from IMERG (blue boxes in c). The boxes extend from the lower to the upper quartile with the middle line depicting the median. The whiskers mark the lower and upper 2.5 percentiles. The $H_2O$ (evaluated using the natural logarithm), $\delta D$ and precipitation distributions are given as difference to the corresponding instrument-specific average over the full available period (see Table 1).

average and the main data spread lays between $-100 - 50\,‰$. In addition, strong outliers become apparent for IASI in $\delta D$ with values far below $-100\,‰$, but not in $H_2O$, which instead remains in an overall compact and slightly increased data range. This can be understood in the context of the discussion in Sect. 3.1, where convective systems were found to create marked Super-Rayleigh signals through processes like rain evaporation and equilibration, with the effect to reduce $\delta D$ only. For 2020,

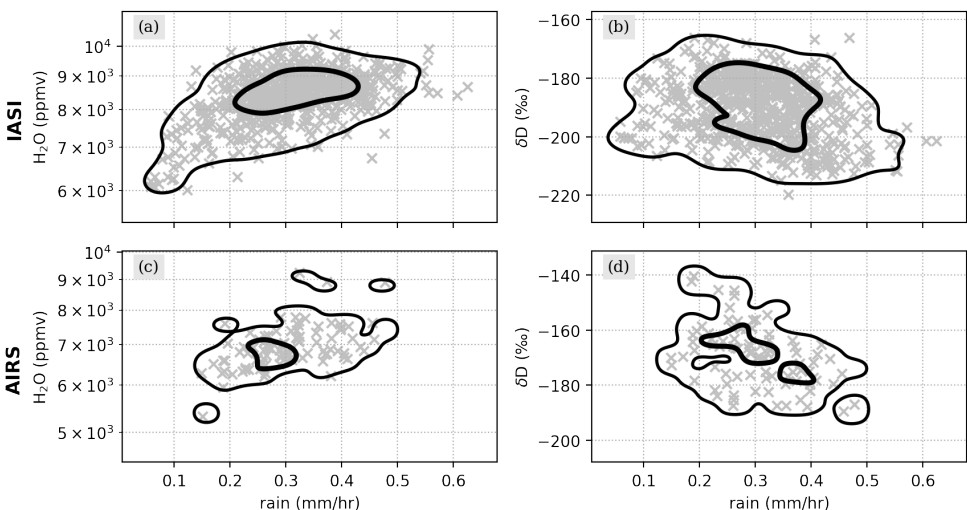

**Figure 13.** Comparison of the $H_2O$ and $\delta D$ data from IASI and AIRS against IMERG precipitation data. For this purpose, the data from Fig. 10 have been re-gridded onto a regular $1° \times 1°$ grid for the comparison of IASI against IMERG and onto a regular $2.5° \times 2.5°$ for the comparison of AIRS against IMERG. The re-gridding has been achieved by averaging all satellite data within a grid cell. Then, the compared data are displayed as scatter and described as two-dimensional histogram contours (indicating the distribution of the main 95 % and 50 % samples).

IASI again indicates relatively strong depleted $\delta D$ signals (median below -10‰ and minimum values between -100 to -75‰), but interestingly also $H_2O$ exhibits comparably low minimum values, while its median value is aligned with the full period average. This can be explained through the corresponding precipitation signals, which similarly show a median value close to the full period median, but also a large extent in the observed data range, reaching from around $-0.3\,\text{mm}^{-1}$ to over $0.4\,\text{mm}^{-1}$.

This suggests that 2020 was characterized by both intense convective and dry intrusion events, with the former resulting in humid and depleted and the latter in dry and depleted air masses.

The comparison of the different years underlines that also an anti-correlation between precipitation amount and $\delta D$ in the vapour phase can be observed. This becomes particularly evident by contrasting individual $H_2O$ and $\delta D$ data points against the corresponding precipitation data, as shown in Fig. 13. While $H_2O$ increases for intensifying precipitation, we again observe

the aforementioned anti-correlated relation between precipitation rates and $\delta D$, with higher precipitation rates corresponding to lower $\delta D$. This effect has also been found in collected rain water samples (e.g. Dansgaard, 1964; Lee and Fung, 2008; Risi et al., 2008a; Tharammal et al., 2017) as well as to water vapour $\delta D$ values (Lawrence et al., 2004; Worden et al., 2007; Tremoy et al., 2012, e.g.), the so-called *amount effect*. Various mechanisms have been proposed for explaining this anti-correlation observed in the vapour phase, such as dry mixing due to convective (Risi et al., 2008a) and meso-scale (Kurita, 2013) downdrafts as

well as the occurrence of partial rain evaporation (e.g. Worden et al., 2007; Risi et al., 2010b; Noone, 2012), which all have been identified to be pronounced within deep convection (Lacour et al., 2018). In this way, this is consistent with the observed

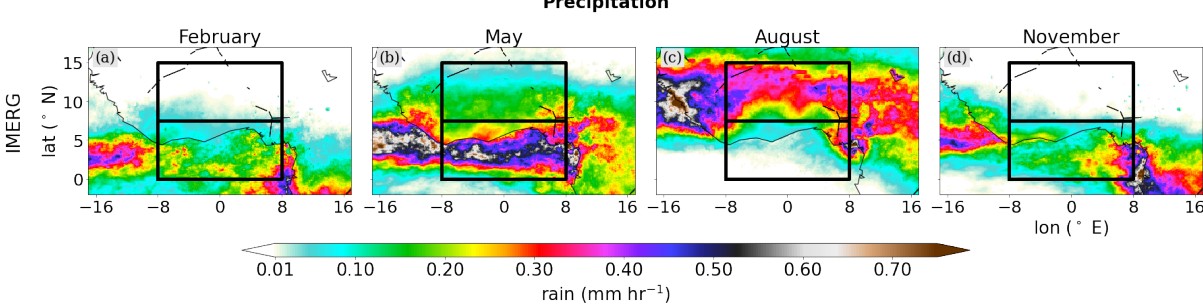

**Figure 14.** Horizontal distributions of monthly averaged precipitation from IMERG over West Africa on the native 0.1° grid. The black rectangles denote the chosen target regions for the Guinea Coast (0°–7.5° N) and the Sahel (7.5°–15° N).

anti-correlation between $H_2O$ and $\delta D$ over the Sahel, which, as discussed in Sect.3.1, was found to be the result of intense rain-vapour interactions associated with convective activity.

Despite differences in the considered time frames, the analysis of Fig. 12 with respect to the decomposition of $\delta D$ anomalies shown in Fig. 11 provides further insights in this context. Following the discussion of the latter in Sect. 3.3, the decomposition method has revealed that the main driver for year-to-year anomalies in $\delta D$ are data samples that have experienced significant convective events, i.e. the post-rain events. As the overall fraction of post-rain air masses has been stable over the considered period (see Fig. 11a), the conclusion was made that the variations in $\delta D_{\text{post-rain}}$ may be a result of varying intensity of the convective events. In particular for the years 2018 and 2019, the IASI $\delta D_{\text{post-rain}}$ anomalies reach minimum values and are substantially lower than for 2015 and 2016. Since 2018 and 2019 were associated with stronger precipitation peaks than 2015 and 2016 (see Fig. 12), this suggests that stronger precipitation rates account for the negative $\delta D_{\text{post-rain}}$ in Fig. 11 and, hence, for the drop in $\delta D$ for 2018 – 2020 as observed in Fig. 12. However, these results should be treated with a degree of caution, as Fig. 11 refers to observations from June – July and Fig. 12 from August.

In conclusion, this section shows that the anti-correlated relation between $H_2O$ and $\delta D$ in mid-tropospheric water vapour as a result of Sahelian convection forms a stable signal during the monsoon period . Moreover, we find that the degree of depletion in $\delta D$ relates to the formation of precipitation events in this region.

## 5   Mean seasonal cycle of {$H_2O$, $\delta D$} signals

In a third step, we put the seasonal WAM development into a climatological context. The aim is to investigate to which extent the change in circulation and monsoon convection affect the isotopic signature of the tropospheric water vapour over the year and whether further control processes may be identified in the {$H_2O$, $\delta D$} phase space.

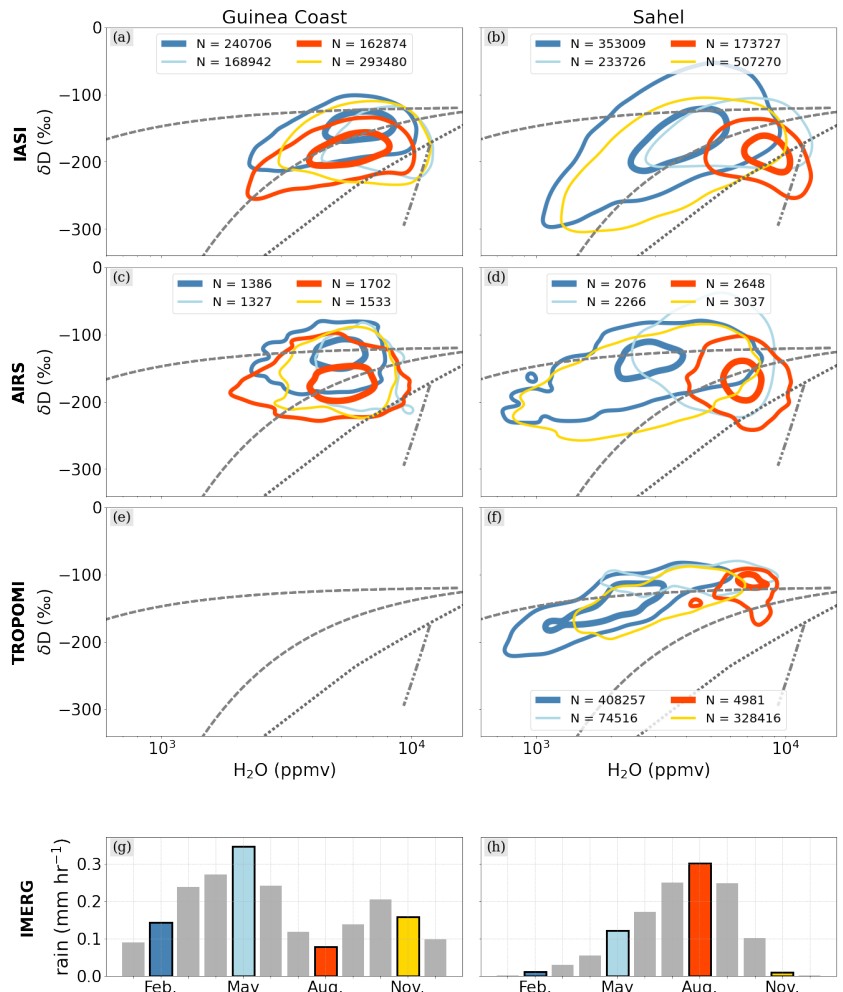

**Figure 15.** First to third row: {$H_2O$, $\delta D$} pair distributions from IASI and AIRS for the Guinea Coast (a, c) and from IASI, AIRS and TROPOMI for the Sahel (b and d), averaged for February, May, August and November from the available periods, respectively (see Table 1). The contours indicate the distribution of the main 95 % and 50 % of the respective data points and with the corresponding total data numbers given in the legend. Bottom row: Monthly precipitation averaged over the Guinea Coast and the Sahel. The color-coded bars correspond to the time-periods of the respective contours in (a) and (b).

## 5.1 Seasonality of precipitation

The subject of our discussion is averaged data from February, May, August and November, with these months proving useful of tracking the overall different seasonal cycles in precipitation over the Guinea Coast and the Sahel. Bringing together the evolution of the horizontal precipitation distributions (Fig. 14) and the region-specific precipitation histograms (Fig. 15g, h), 450   we can characterize the selected months as follows:

- During February, when dry Harmattan winds penetrate for south into the Guinea Coastal region, the land mass of West Africa (Sahel) is mostly rain-free and precipitation is largely restricted to a weak zonal band over the Tropical Atlantic (Guinea Coast).

- In May, the precipitation intensifies strongly, though still remains over the ocean. As a result, the Guinea Coast experiences a first rain season and first precipitation events occur over the Sahel.

- In August, the monsoon is fully established and strong southerly winds from the Tropical Atlantic push the precipitation band northwards into the Sahel. While a little dry season develops over the Guinea Coast, the Sahel shows its annual precipitation peak.

- During November, the monsoon has retreated, leading to a distribution largely similar to February. The Sahel is overall dry and the southwards propagation of the precipitation band creates a second, weaker rain season over the Guinea Coast.

In this way, we confirm that the IMERG precipitation products of the selected months serve as a valuable proxy for tracking the mean dynamical cycle over the West Africa. 154).

## 5.2 Seasonality of isotopic signals

### Guinea Coast

Throughout the whole year, the Guinea Coast shows relatively high and overall stable signals in $H_2O$ and $\delta D$ for both IASI and AIRS (Fig. 15a, c; TROPOMI is not considered for the target domain over the Guinea Coast), and the corresponding $\{H_2O, \delta D\}$ pair distributions gather around the moist and enriched hyperbolic mixing regime. This is the result of the Tropical Atlantic acting as strong evaporative source of moisture into the troposphere (Frankenberg et al., 2009; Diekmann et al., 2021a). The rain seasons during May and November further moisten the already humid troposphere (up to appprox. 10 000 ppmv) and additionally lead to a slight depletion in $\delta D$ (down to approx. –200 ‰). As a result to the increased occurrence of rain-vapour interaction during the rain seasons, where in particular rain condensation and evaporation deplete the water vapour in $\delta D$, the respective $\{H_2O, \delta D\}$ pair distributions depict a minor shift from the mixing regime to the Rayleigh curve and even slightly to the Super-Rayleigh regime. During the dry season between the two rain seasons, the $\{H_2O, \delta D\}$ pairs show a stretch along the indicated mixing curve down to 2000 ppmv in $H_2O$ and –230 ‰ in $\delta D$, assumingly as response to dry intrusions mixing into the Guinean troposphere.

In contrast to the $\{H_2O, \delta D\}$ evolution over the Guinea Coast, drastic differences become evident between the dry winter and rainy summer periods for the study domain over the Sahel, which can be understood in the context of the discussion about the seasonal movement of the precipitation belt (Sect. 5.1). Due to its more northern position, the Sahelian troposphere experiences a stronger dehydration than the Guinea Coast during the dry season (represented here by November and February). With the precipitation maximum being located over the Tropical Atlantic, almost no precipitation occurs over the Sahel. Further, the descending branch of the Hadley cell is found to feed dry and depleted air masses into the North and West African troposphere (Frankenberg et al., 2009), which are then transported by the far-south reaching Saharan Harmattan winds towards

the Sahel. With only weak evaporation taking place, this creates a strong mixing signal in the {$H_2O$, $\delta D$} distribution of the Sahelian troposphere reaching minimum values of 1000 ppmv for $H_2O$ and –300‰ for $\delta D$ for IASI. This is in agreement with Diekmann et al. (2021a), where this moist mixing curve was found to well represent the near-surface moistening along the Harmattan winds and the Atlantic westerly inflow. Similar to the findings over the Guinea Coast, AIRS depicts a similar spread in $H_2O$, while for $\delta D$ an enriched bias in low $\delta D$ regimes is documented. This reduces the observed spread in $\delta D$, which varies between –200 and –100‰ over the full $H_2O$ range. This means that for dry conditions the AIRS data follow significantly less a mixing line if compared to IASI. We assume that this can be explained by differences in the $H_2O$ and $\delta D$ sensitivities for dry conditions. Typically, retrieval results for $\delta D$ have much lower sensitivity than for $H_2O$, i.e. the paired {$H_2O$, $\delta D$} data product captures well the decrease in $H_2O$ with altitude, but falls short in accurately depicting the decrease in $\delta D$ (see also Section 7 in Schneider et al. (2016)). The {$H_2O$, $\delta D$} harmonization as part of the MUSICA IASI post-processing (see Section 2.1.1) serves to increase the sensitivity of the $\delta D$ retrieval to dry conditions and thus ensures very similar sensitivities of $H_2O$ and $\delta D$, with the effect that the respective IASI {$H_2O$, $\delta D$} pairs are distributed close to the mixing line.

For TROPOMI, the column-averaged $\delta D$ represents mainly the $\delta D$ values of the humid boundary layer (see discussion in Sect. 2.1.1) and there is only a weak variability, with a data range between -100 and -200‰. The column-averaged $H_2O$ can be interpreted as a measure of the depth of the humid near-surface layer. For a deep humid layer, as it is the case in the monsoon layer over West Africa during the rain season (Cornforth et al., 2017), $H_2O$ values are much higher than for a shallow boundary layer during the dry season. These particularities of the column-averaged {$H_2O$, $\delta D$} pair data create strong signals along the mixing line that arise from the mixing of a humid and a very dry end member (column-averaged data represent a vertically completely mixed atmosphere).

**Sahel**

Even though the Sahelian winter is mostly dry and rain-free, rare but heavy precipitation events may form as response to strong tropical-extratropical interactions between the upper tropospheric wind regimes, the monsoon southerlies and the Saharan Harmattan winds (e.g. Knippertz and Martin, 2005; Knippertz and Fink, 2009; Davis et al., 2013). Occasional northward shifts of the moist southerly winds allow the precipitation belt from the Gulf of Guinea to reach (for boreal winter) unusually high latitudes. This explains the occurrences of high moisture contents even in the dry winter season (up to 10000 ppmv), ultimately resulting in a large variability in the corresponding {$H_2O$, $\delta D$} pair distributions around the mixing lines for all three satellite products. As has been discussed for the Guinea Coast, we again observe discrepancies in $\delta D$ for dry conditions between IASI and AIRS. While IASI shows clear mixing signals reaching to 1000 ppmv and –300‰, AIRS appears less depleted in $\delta D$ for the same $H_2O$ regime, which is assumed to be a result of the optimization of IASI $H_2O$ and $\delta D$ retrieval states for dry conditions (see 2.1.1).

The winter distribution of Sahelian {$H_2O$, $\delta D$} pairs reveals another interesting feature in the $H_2O$ range of 2000–8000 ppmv. The corresponding contour exhibits enriched $\delta D$ values of larger than -100 ‰, i.e. above the shown process curves. This is likely caused by vertical mixing of the very dry sub-tropical free tropospheric air with more humid air of the Saharan boundary layer. This mixing peaks in the dust loaded Saharan air layer. Respective high $\delta D$ values for this layer have been reported in several studies and using in-situ measurements from ground and aircraft (Dyroff et al., 2015; González et al., 2016), using data

from ground- and space-based remote sensing (Schneider et al., 2015) and using isotopic-enabled modeling (Dahinden et al., 2023).

As discussed in Sect. 5.1, the monsoon onset is characterised by the shift of maximum precipitation from the Guinea Coast to the Sahel during summer. The precipitation peak over the Sahel during August goes along with a strong increase in $H_2O$ (data range between 5000–10 000 ppmv), whereas $\delta D$ shows decreasing tendencies (below –200‰) and points to the previously observed anti-correlation between $H_2O$ and $\delta D$. Figure 15 shows that the enhanced depletion during summer is strongly associated with signals below the Rayleigh prediction and creates {$H_2O$, $\delta D$} structures following the Super-Rayleigh line

indicated in the figure. These features are also detected by the coarser resolved AIRS dataset, where the anti-correlation is less pronounced but still observable. Furthermore, despite the vertical smoothing over the troposphere for TROPOMI, which leads to a shift of the {$H_2O$, $\delta D$} pair contour to higher $\delta D$ and lower $H_2O$, the Super-Rayleigh structures stand out as distinct spike in the otherwise round {$H_2O$, $\delta D$} pair contour. This underlines the strong depleting effect of the rain-vapour interactions that mainly occur in a rather shallow layer in the free troposphere (Diekmann et al., 2021a).

In conclusion, we find that the {$H_2O$, $\delta D$} data over the Sahel reflect the overall transition from the dry winter season, which is mostly governed by dry air mass mixing, to the summer monsoon season, when moist air reaches into the Sahel and drives microphysical processes related to convection. In contrast, the Guinea Coast develops a comparably less pronounced seasonality in the {$H_2O$, $\delta D$} phase space due to its structurally different meteorological conditions. In this way, this section demonstrates that the {$H_2O$, $\delta D$} phase space is capable of tracking individual effects from dynamical and microphysical processes that

control the atmospheric state over West Africa.

## 6 Conclusion and Outlook

The goal of this study was to characterize the joint variability of $H_2O$ and $\delta D$ in the West African troposphere using space-based remote sensing data and to shed light on the key processes controlling this variability. For this purpose, global and multi-annual data from the state-of-the-art infrared satellite sensors IASI, AIRS and TROPOMI served to track pairs of tropospheric $H_2O$

and $\delta D$ abundances over the Guinea Coastal region and the Sahel from the interannual down to the convective scale. Based on IMERG precipitation estimates, we derive a clustering method to distinguish observations affected by convection or not. Further, the use of idealized process curves in {$H_2O$, $\delta D$} phase space, as derived by the Lagrangian process attribution study in Diekmann et al. (2021a), allowed to interpret the remotely sensed {$H_2O$, $\delta D$} pairs with respect to the governing dynamical and microphysical processes.

By bringing together the isotopic and precipitation datasets for the Sahel and the Guinea Coast, we were able to derive the following conclusions:

- Despite the rather simple concept of our precipitation-based clustering method, we find that convection from different regions lead to different structures in {$H_2O$, $\delta D$} phase space during the monsoon: Over the Sahel, where convective squall line events are opposed to strong dry air intrusions, convective precipitation is associated with the coupling of rain

condensation and evaporation. As a result, this leads to a moistening of the mid-troposphere as well as to an enhanced

depletion in $\delta D$, creating clearly anti-correlated features in the $\{H_2O, \delta D\}$ phase space towards the Super-Rayleigh regime. In contrast, the Guinea Coast is less affected by dry intrusions, thus here convective precipitation goes along with overall moist and enriched air masses.

– The anti-correlated behaviour between $H_2O$ and $\delta D$ over the Sahel during the monsoon peak remains a robust feature in the multi-year perspective. In addition, an anti-correlation between precipitation amount and $\delta D$ becomes apparent, where stronger precipitation events are associated with more enhanced $\delta D$ depletion, thus pointing towards the amount effect in water vapour.

– The analysis of the mean seasonal cycles demonstrates that the anti-correlated relation between $H_2O$ and $\delta D$ is overall limited to the monsoon period and to the Sahel. During the Sahelian winter, which is overall dry and rain-free, predominant mixing signals form as response to strong dynamical processes over West Africa. Again in contrast, the Guinea Coast appears consistently moist and enriched and due to the missing dry air intrusions no significant depletion in $\delta D$ develops.

The discussed features in $\{H_2O, \delta D\}$ over the Guinea Coast and the Sahel are particularly pronounced and qualitatively consistent for the vertically resolved water vapour isotope products from IASI and AIRS. Compared to AIRS, IASI reveals a difference of approx. $1000 - 1500$ ppmv in $H_2O$ and of approx. $10 - 20\text{\textperthousand}$ in $\delta D$. In contrast, the total-column-averaged TROPOMI product differs markedly from IASI and AIRS. Whereas the seasonal TROPOMI $\{H_2O, \delta D\}$ cycle is similar to the signals observed by the other instruments, the anti-correlation between $H_2O$ and $\delta D$ associated to Sahelian convection is less evident due to weak $\delta D$ variability. These differences are due to the fact that the TROPOMI $H_2O$ data mainly indicate the depth of the monsoon layer (i.e. high total-column-average $H_2O$ referring to a deep monsoon layer), whereas its $\delta D$ reflects the isotopic composition predominantly from within the monsoon layer.

In conclusion, this study underlines the value of utilizing paired distributions of $H_2O$ and $\delta D$ observed from space to study the atmospheric water cycle in terms of tropical convection, microphysical processes and the large-scale circulation. Together with new frameworks to describe isotopic processes using satellite data in a more quantitative way (e.g. Galewsky et al., 2023), with new methods to combine retrieval results from different instruments in order to create synergy products with increased information content (e.g. for IASI and TROPOMI, see Schneider et al., 2022) and with new techniques to assimilate isotopic observations into atmospheric models (Toride et al., 2021; Schneider et al., 2023), promising opportunities are emerging to further exploit the process-based view on paired $\{H_2O, \delta D\}$ data from space and, hence, to reveal new insights into atmospheric processes and their numerical representations.

*Data availability.* The MUSICA IASI data are available at https://doi.org/10.35097/415. The AIRS data can be accessed at https://tropess. gesdisc.eosdis.nasa.gov/data/TROPESS_Reanalysis_Summary/TRPSYL2HDOAIRSORS.1/. Information on how to access the TROPOMI data are documented in https://s5pinnovationh2o-iso.le.ac.uk/elementor-129/. The GPM IMERG data are available at https://gpm.nasa.gov/data/directory.

*Author contributions.* Matthias Schneider, Benjamin Ertl and Christopher Diekmann have created and provided the MUSICA IASI data set, whereby they have been supported by Farahnaz Khosrawi and Frank Hase. Hartmut Boesch and Tim Trent have created and provided the Sentinel-5P data set with support by Amelie Ninja Roehling. John Worden has created and provided the AIRS dataset. Peter Knippertz has provided the scientific background about the West African Monsoon. Peter Knippertz, Matthias Schneider and Christopher Diekmann have designed the concept of the study. All authors supported the generation of the final version of this manuscript.

*Competing interests.* Farahnaz Khosrawi is a member of the editorial board of Atmospheric Chemistry and Physics.

*Acknowledgements.* This work has been financially supported in the context of the projects MOTIV and TEDDY (funded by the Deutsche Forschungsgemeinschaft under project IDs/Geschäftszeichen 950290612604/GZ:SCHN1126/2-1 and 416767181/GZ:SCHN1126/5-1, respectively) and the project Sentinel-5P+Innovation H2O-ISO (funded by the European Space Agency, ESA Contract No. 4000127561/19/I-NS). Further, it has benefited from the project MUSICA (funded by the European Research Council under the European Communitys Seventh Framework Programme (FP7/2007-2013)/ERC Grant Agreement number 256961). The technical processing for the presented results has been performed using the supercomputer HoreKa, which is funded by the Ministry of Science, Research and the Arts Baden-Württemberg and by the German Federal Ministry of Education and Research. Part of this research was carried out at the Jet Propulsion Laboratory, California Institute of Technology, under a contract with the National Aeronautics and Space Administration.

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
