# Peer review of "Water vapour isotopes over West Africa as observed from space: which processes control tropospheric $H_2O/HDO$ pair distributions?"

_EGUsphere, 2024_

## Referee Comment (RC1)

**Review of Diekmann et al**

**September 9, 2024**

This article documents variations in the tropospheric water vapor isotopic composition over West Africa at the convective, seasonal and inter-annual scales, using 3 products of satellite observations: IASI, AIRS and TROPOMI. Processes responsible for the isotopic variations are analyzed. This paper is consistent with many previous papers showing the impact of convective processes and air mass mixing on the isotopic composition of water vapor. The article is not exceptionally novel, but contributes to a more confident understanding of processes controlling the isotopic compositon of water vapor.

The article is well written and well illustrated. My comments are minor.

**1 General comment**

What is the added value of $\{H_2O, \delta D\}$ pairs? Can they teach us anything new? The paper explains well how $\{H_2O, \delta D\}$ are consistent with processes that we already know. But it doesn't show how they could improve the konwledge. It's fine, maybe they just cannot improve the knowledge. But then maybe some sentences would need to be toned down. e.g. l 3: " "quantify the atmospheric branches""; l 20: "underlines the overall value..."; l 471-473: "underlines the strong potential".

**2 Detailed comments**

- l 3: "quantify the atmospheric branches": unclear. We can quantify some fluxes or reservoirs, or the contribution of some processes, but not clear what it means to quantify a branch.

- l 8-9: "from a convective as well as seasonal perspective and with respect to"-> "at the convective and seasonal scales and"

- l 12: "$\delta D$ depletion" -> "$\delta D$ depletion in the vapor"

- l 13: remove ""without showing significant $\delta D$ depletion": obvious after "enriched signals"

- l 15: "anti-correlation": at what time scale? How different is (3) different from (1)?

- l 30: what does "degree of dessication" means? Could this be clarified for people from the isotopic community who don't know about it?

- l 39 and around: is convection really the main source of uncertainty in climate projections over the Sahel? Recent studies suggest a key role of Atlantic SSTs as well, e.g. [Monerie et al., 2023]

- l 63: "in both liquid hydrometeors and ambiant vapor" -> "in ambiant vapor". Evaporation rather enriches the droplets.

- l 87: "interannual seasonal -> "interanual".

- l 182: clarify the average: average over the previous day? l 184: why do we need half-hourly IMERG observations rather than just daily? Are air masses followed along back-trajectories to calculate average rainfall? Or is the rainfall local?

- Fig 2: give some more context for these lines, e.g. in the caption. What difference between the 2 mixing lines? How are initial conditions and end members chosen?

- l 252: you may cite [Risi et al., 2021] to support the impact of snow melt on the water vapor composition.

- l 273-376: If I understand well, you interpret the absence of isotopic difference between non-rain and post-rain events by the compensation at lowest levels of the enrichment by surface evaporation and depletion by convection? If the case, how can you explain the depletion of the vapor observed after rain events in surface observations [Tremoy et al., 2012, Tremoy et al., 2014]?
  Actually, fig 6 shows that the humidity is not even larger for post-rain events. This questions whether the clustering methods applied to TROPOMI observation is really comparing non-rain and post-rain events. Is it possible that near the surface, the water vapor recovers more quickly after the event due to surface evaporation, and so the clustering methods based on average rain over the previous days might not properly capture post-rain vapor?

- l 278 and around: I'm not sure about this rationale. The depleting effect actually accumulates along the descent in unsaturated and mesoscale downdrafts of convective systems, as shown by cloud-resolving simulation [Torri, 2022, Risi et al., 2023]. So we do expect, and generally observe, depletion near the surface after convective systems in the Sahel.

- Fig 6: this compares non-rain and post-rain events. It would have been interesting to document the impact of the intensity of rain events, e.g. through the rainfall rate. If this is too much work for this article, this could be mentionned as a perspective.

- l 286 and next lines: "types of convection": the impact of convection type was not addressed in this study. Only non-rain and post-rain conditions are compared. To analyze the role of convection type, a more sophisticated clustering method would be useful, e.g. squall line vs isolated systems. This paragraph up to l 296 needs to be completely revised.
  It would have been interesting to link the $\delta D$ to convection type. If this is too much work for this article, this could be mentionned as a perspective.

- l 295: "unaggregated convection (as is the case in Sahelian squall lines)": this is the contrary! Squall lines are highly aggregated convective systems e.g. [Abramian et al., 2022]. Aggregated means that convection is gathered into one big system, whereas unaggregated means that convection is scattered into several isolated systems, e.g. [Bretherton et al., 2005, Tobin et al., 2012]

- l 311: I can see only a few ‰ drop in fig 7.

- Fig 7: clarify in the caption what the error bars mean. Is it the standard deviation of all instantaneous values?

- Interpretation of fig 7:

  - To better see the link between rainfall and $\delta D$, could a scatter plot of $\delta D$ vs rainfall anomalies be added?

  - Why are more rainy years more depleted? Is it because there are more rainy events, which are more depleted (fig 6)? Or is it because non rainy events are more depleted? Or because rainy events are more depleted, e.g. because they are more intense? To answer this question, it could be easy to link Fig 7 to fig 6 with a decomposition method: $\Delta\delta D = \Delta r \cdot (\delta D_{rain} - \delta D_{norain}) + r \cdot \Delta\delta D_{rain} + (1 - r) \cdot \Delta\delta D_{norain}$, where $\Delta\delta D$ is the anomaly between high and low rainfall years and $r$ is the fraction of rain samples in the yearly average.

- Around l 365: is it possible that the smaller sensitivity of AIRS could be due to the larger impact of the a-priori profile on AIRS than on IASI, i.e. smaller sensitivity?

- l 384: "As result" -> "As a result"

- Somewhere: the recent study by [Dahinden et al., 2023] would deserve to be cited.

**References**

[Abramian et al., 2022] Abramian, S., Muller, C., and Risi, C. (2022). Shear-convection interactions and orientation of tropical squall lines. *Geophys. Res. Lett.*, 49(1):e2021GL095184, DOI: doi.org/10.1029/2021GL095184.

[Bretherton et al., 2005] Bretherton, C. S., Blossey, P. N., and Khairoutdinov, M. (2005). An Energy-Balance Analysis of Deep Convective Self-Aggregation above Uniform SST. *J. Atmos. Sci.*, 62:4273–4292.

[Dahinden et al., 2023] Dahinden, F., Aemisegger, F., Wernli, H., and Pfahl, S. (2023). Unravelling the transport of moisture into the saharan air layer using passive tracers and isotopes. *Atmospheric Science Letters*, 24(12):e1187.

[Monerie et al., 2023] Monerie, P.-a., Biasutti, M., Mignot, J., Mohino, E., Pohl, B., and Zappa, G. (2023). Storylines of sahel precipitation change: Roles of the north atlantic and euro-mediterranean temperature. *Journal of Geophysical Research: Atmospheres*, 128(16):e2023JD038712.

[Risi et al., 2021] Risi, C., Muller, C., and Blossey, P. (2021). Rain evaporation, snow melt, and entrainment at the heart of water vapor isotopic variations in the tropical troposphere, according to large-eddy simulations and a two-column model. *J. Adv. Model. Earth Sci.*, 13(4):e2020MS002381, DOI: https://doi.org/10.1029/2020MS002381.

[Risi et al., 2023] Risi, C., Muller, C., Vimeux, F., Blossey P NBlossey P N, G., Dufaux, C., and Abramian, S. (2023). What controls the mesoscale variations in water isotopic composition within tropical cyclones and squall lines? cloud resolving model simulations. *J. Adv. Model. Earth Sci.*, 15 (4):e2022MS003331.

[Tobin et al., 2012] Tobin, I., Bony, S., and Roca, R. (2012). Observational evidence for relationships between the degree of aggregation of deep convection, water vapor, surface fluxes and radiation. *J. Clim.*, 25(20):6885–6904 , DOI: https://doi.org/10.1175/JCLI–D–11–00258.1.

[Torri, 2022] Torri, G. (2022). Isotopic equilibration in convective downdrafts. *Geophys. Res. Lett.*, 49(15):e2022GL098743.

[Tremoy et al., 2012] Tremoy, G., Vimeux, F., Mayaki, S., Souley, I., Cattani, O., Favreau, G., and Oi, M. (2012). A 1-year long delta18O record of water vapor in Niamey (Niger) reveals insightful atmospheric processes at different timescales. *Geophys. Res. Let.*, 39(8):DOI: https://doi.org/10.1029/2012GL051298.

[Tremoy et al., 2014] Tremoy, G., Vimeux, F., Soumana, S., Souley, I., Risi, C., Cattani, O., Favreau, G., and Oi, M. (2014). Clustering mesoscale convective systems with laser-based water vapor delta18O monitoring in Niamey (Niger). *J. Geophys. Res.*, 119(9):5079–5103, DOI: 10.1002/2013JD020968.

---

## Author Response (AR1)

**Author Response to the Referee Comments to the manuscript "Water vapour isotopes over West Africa as observed from space: which processes control tropospheric H2O/HDO pair distributions?" [EGUSPHERE-2024-1613] submitted to Atmospheric Chemistry and Physics.**

We would kindly thank the anonymous referee for providing a review of the manuscript. The individual comments are listed below (shown in red) including our responses (shown in black) and the changes applied to the manuscript (shown in black italics).

"This article documents variations in the tropospheric water vapor isotopic composition over West Africa at the convective, seasonal and inter-annual scales, using 3 products of satellite observations: IASI, AIRS and TROPOMI. Processes respconsible for the isotopic variations are analyzed. This paper is consistent with many previous papers showing the impact of convective processes and air mass mixing on the isotopic composition of water vapor. The article is not exceptionally novel, but contributes to a more confident understanding of processes controlling the isotopic compositon of water vapor.

The article is well written and well illustrated. My comments are minor."

Thank you very much for the valuable and insightful feedback!

**General Comment**

"What is the added value of {H2O, δD} pairs? Can they teach us anything new? The paper explains well how {H2O, δD} are consistent with processes that we already know. But it doesn't show how they could improve the konwledge. It's fine, maybe they just cannot improve the knowledge. But then maybe some sentences would need to b e toned down. e.g. l3: "quantify the atmospheric branches"; l20: "underlines the overall value…"; l 471-473: "underlines the strong potential"."

As has been demonstrated in previous studies, the added value of {H2O, δD} pairs is that it can shed light onto the impact of isotopic processes in a way that would not be possible by individual H2O or δD distributions alone. The comment is correctly assuming that our study does not increase our knowledge about the isopotic processes per se. Instead, it shows that various processes can be detected on different time scales using space-based datasets of H2O and δD, and, hence, fosters our general understanding of {H2O, δD} pair variability from an observational point of view.

Therefore, we understand that we will need to tone down some phrases and focus more on the actual scope and outcomes of this study.

*L3. "… quantify the atmospheric branches …" → "…make it difficult to disentangle the contributions of individual atmospheric processes to the hydrological cycle in observations and models." (see comment below)*

*L.20 "… underlines the overall value …" → "This study is the first to apply the new and comprehensive isotopic datasets of IASI, TROPOMI and AIRS to the WAM. Despite the large complexity of the monsoon system, it confirms that satellite-based H2O and dD data are useful to detect the impact of different microphysical and dynamical processes in the isotopic composition of water vapour at the seasonal and convective scales."*

*L.471-473 "… underlines the strong potential … to foster our process understanding …" –> "In conclusion, this study underlines the value of utilizing paired distributions of H2O and δD observed from space to study the atmospheric water cycle in terms of tropical convection, microphysical processes and the large-scale circulation."*

**Detailed Comments**

"l 3: "quantify the atmospheric branches": unclear. We can quantify some fluxes or reservoirs, or the contribution of some processes, but not clear what it means to quantify a branch."

Thank you for pointing out this inaccurate wording. The intended meaning of this phrase was to refer to the challenge of assessing the individual impact of atmospheric processes within the hydrological cycle in observations and models. We will apply the necessary adjustments to this phrase:

*"..make it difficult to disentangle the contributions of individual atmospheric processes to the hydrological cycle in observations and models."*

"l 8-9: "from a convective as well as seasonal perspective and with respect to" -> "at the convective and seasonal scales and""

*Ok, we will rephrase it.*

"l 12: "δD depletion -> δD depletion in the vapor""

*Ok, will be corrected.*

"l 13: remove "without showing significant δD depletion": obvious after "enriched signals""

Ok, will be removed.

"l 15: "anti-correlation": at what time scale? How different is (3) different from (1)?"

(3) refers to the anti-correlation between precipitation amount and δD, while (1) refers to the anti-correlation between H2O and δD. For (3), we observe the described effect on monthly averages compared for different years. We will adjust the description of (3) accordingly to underline the considered time scale.

*"During the Sahelian monsoon peak, an anti-correlation between increasing precipitation amount and decreasing δD becomes apparent in a year-to-year comparison"*

"l 30: what does "degree of dessication" means? Could this be clarified for people from the isotopic community who don't know about it?"

In the context of Africa, the term of desiccation refers to the progressive drying of soil and the subsequent desertification, e.g. as result of decreasing rainfalls and intensifying droughts. We will adjust this phrase in the manuscript accordingly to clarify its meaning:

*"for the degree of desiccation, i.e. the progressive soil drying and subsequent desertification as result of decreasing rainfalls and intensifying droughts"*

"l 39 and around: is convection really the main source of uncertainty in climate projections over the Sahel? Re ent studies suggest a key role of Atlantic SSTs as well, e.g. [Monerie et al., 2023]"

Thanks for pointing this out. We agree that indeed there are further important factors driving the uncairtenty in climate predictions over the Sahel. We will tone down the term "major" to "one of the main" and refer to Atlantic SSTs as example for further factors:

*"Studies have shown that, besides other factors such as changes in Atlantic sea surface temperatures (e.g. Monerie et al., 2023) one of the main sources of uncertainty in climate projections over the Sahel is convection,"*

"l 63: "in both liquid hydrometeors and ambiant vapor" -> "in ambiant vapor". Evaporation rather enriches the droplets"

Ok, we will rephrase it.

"l 87: "interannual seasonal -> "interanual"."

Ok, we will rephrase it.

"l 182: clarify the average: average over the previous day?

Yes, the term "average" refers to the averaged precipitation of the previous day. Considering the response to the next comment, we will rephrase the corresponding sentence as follows:

*"If within a chosen grid box and at a given calendar day the daily precipitation summed over the previous three days remains below a chosen threshold ..."*

l 184: why do we need half-hourly IMERG observations rather than just daily? Are air masses followed along back-tra je tories to calculate average rainfall? Or is the rainfall local?"

We agree with the reviewers comment that the description of the clustering method is somewhat confusing.
We have to add the comment that we actually utilize daily IMERG observations instead of hourly to run the precipitation clustering method. Following the IMERG data description, the daily IMERG observations are derived from averaging all corresponding half-hourly IMERG observations at the considered 0.1 x 0.1 degree grid. Based on this, the daily IMERG observations are distributed via the data provider as daily precipitation (i.e. mm per day). As we make direct use of these daily precipitation products, no further averaging is required for identifying post-rain events with the considered clustering method. We added following sentence to the data description:

*"The daily precipitation product is derived from averaging half-hourly precipitation rates at the considered $0.1° ×0.1°$ grid and afterwards multiplying the mean precipitation rates (in mm per hour) by 24 to obtain daily rates (i.e. mm per day) (Huffman et al. 2019)"*

Further, we agree the consideration of air mass backward-trajectories would be indeed interesting to consider for improving the detection mechanism of non-rain and post-rain events. As this exceeds the scope of this study, we will add it as perspective.

*"Despite the rather simple nature of this approach (e.g. neglecting effects such as the advection of air masses from adjacent grid boxes, which could be addressed by Lagrangian backward trajectories), the current choice of method and thresholds proves successful to identify the targeted precipitation states,"*

"Fig 2: give some more context for these lines, e.g. in the caption. What difference between the 2 mixing lines? How are initial conditions and end members chosen?"

We will add following descriptions based on Diekmann et al. 2021a into the caption of Fig. 2:

- The air mass mixing curves are representative for dry and moist mixing processes over West Africa, with considering the mixing members x1 = (5e1 ppmv, -700 ‰,) and x2 = (1.53e4 ppmv, -120 ‰,) for the dry mixing curve (upper mixing curve) and x1 = (1e3 ppmv, -450 ‰,) and x2 = (2.2e4 ppmv, -120 ‰,) for the moist mixing curve (lower mixing curve).

- The Rayleigh curve is computed as Rayleigh process with initial conditions of $\delta D_0$ = - 80 ‰, relative humidity of 90 % and $T_0$ = 30 deg. The Super-Raleigh curve has been defined empirically in order to represent signals due to rain evaporation and equilibration as found along backward air trajectories by Diekmann et al. 2021a.

"l 252: you may cite [Risi et al., 2021] to support the impact of snow melt on the water vapor composition"

Thanks for pointing out this study, which we will add as reference here.

"l 273-276: If I understand well, you interpret the absence of isotopic difference between non-rain and post-rain events by the compensation at lowest levels of the enrichment by surface evaporation and depletion by convection? If the case, how an you explain the depletion of the vapor observed after rain events in surface observations [Tremoy et al., 2012, Tremoy et al., 2014]?
l 278 and around: I'm not sure about this rationale. The depleting effect actually accumulates along the descent in unsaturated and mesoscale downdrafts of convective systems, as shown by cloud-resolving simulation [Torri, 2022, Risi et al., 2023]. So we do expect, and generally observe, depletion near the surface after convective systems in the Sahel.
Actually, Fig 6 shows that the humidity is not even larger for post-rain events. This questions whether the clustering methods applied to TROPOMI observation is really comparing non-rain and post-rain events. Is it possible that near the surface, the water vapor recovers more quickly after the event due to surface evaporation, and so the clustering methods based on average rain over the previous days might not properly capture post-rain vapor?

As these three comments point to the same issue, we will hereafter provide a combined answer.

We appreciate this insightful comment and the constructive suggestions. We understand that in the discussion for the TROPOMI data we have underestimated the near-surface depleting effect associated with convection. In addition to the studies referred in the comment, it can be seen in Fig. 3.9 of Lafore et al. (2017) that warming and drying behing squall lines is very strong due to the dry downdrafts in the rear of the squall line.

Therefore, we believe that the proposed explanation of a rapidly recovering near-surface water vapor after the passing of convection sounds reasonable. This would be supported by Fig. 5 and 6, where $H_2O$ and $\delta D$ are overall similar for non-rain and post-rain conditions. A more sophisticated clustering method with higher temporal frequency would be beneficial in order to create better links between individual convective events and collocated TROPOMI data. Therefore, we will adjust the interpretation of these data accordingly and add the potential improvements for the clustering method as perspective.

*"Accordingly, we find that the boxplots of the non-rain and post-rain classes shown in Fig 10, reveal a high level of agreement in terms of their mean values. This is contrary to various studies that have documented significant drying and depletion of near-surface air after the passage of a convective squall line event as result of dry rear-to-front downdrafts (e.g. Tremoy et al., 2012, 2014; Lafore et al., 2017; Torri, 2022; Risi et al., 2023). The resulting warm and dry air near the surface enhances the evapo-transpiration of surface water, which is associated with enrichment in $\delta D$ (Noone, 2012; Worden et al. 2021) and which thus may accelerate the recovery of water vapor after the convective event. The advection of moist air masses due to the monsoon flow from the Tropical Atlantic into the Sahel further foster the recovery of the post-storm near-surface air (Lafore et al., 2017). Therefore, we assume that a rapidly recovering water vapor may hamper the capability of the clustering to capture actual post-rain conditions (e.g. Super-Rayleigh signals that are typical for atmosphere affected by convection), when applying to TROPOMI and defining the post-rain events by means of the precipitation from the previous day. Instead, a higher temporal frequency would be beneficial to create more direct links between individual convective systems and collocated TROPOMI data."*

"Fig 6: this compares non-rain and post-rain events. It would have been interesting to document the impact of the intensity of rain events, e.g. through the rainfall rate. If this is too much work for this article, this could be mentionned as a perspective."

Thank you for this suggestion. The current design of the considered clustering method only foresees to detect samples for the discrete groups of non-rain and post-rain. The consideration of the impact of rainfall intensity on the post-rain event would be a very interesting analysis, which would require

a further evolution of the clustering method, e.g. to define further post-rain events with different rainfall selection criteria. Thus, we will add this as perspective to the discussion of corresponding figure.

*"Thus, a promising perspective to strengthen the value of the clustering results would be to characterize the non-rain and post-rain clusters as function of precipitation rates."*

"l 286 and next lines: "types of convection": the impact of convection type was not addressed in this study. Only non-rain and post-rain onditions are compared. To analyze the role of convection type, a more sophisticated clustering method would be useful, e.g. squall line vs isolated systems. This paragraph up to l 296 needs to be completely revised.
It would have been interesting to link the δD to convection type. If this is too much work for this arti le, this could be mentionned as a perspective

We appreciate this constructive comment and understand that the clustering method for identifying non-rain and post-rain conditions actually does not provide conclusions on δD as result of different types of convection. We observe regional differences in the isotopic signature of the post-rain events between the Guinea Coast and the Sahel, and we agree that further work would be needed in order to investigate to the reasons for these differences and to which extent different types of convection may account for these findings. Therefore, we will revise the corresponding paragraph and add the analysis with respect to convection types as further perspective.

The changes applied with respect to this comment are included in the response of the subsequent comment.

l 295: "unaggregated convection (as is the case in Sahelian squall lines)": this is the contrary! Squall lines are highly aggregated convective systems e.g. [Abramian et al., 2022]. Aggregated means that convection is gathered into one big system, whereas unaggregated means that convection is scattered into several isolated systems, e.g. [Bretherton et al., 2005, Tobin et al., 2012]"

Thank you for the clarification and correction. We will update the interpretation with respect to the study of Galewsky et al. (2023) as follows:

*"Maranan et al. (2018) have found that convection over Southern West Africa may form in various types, ranging from highly organized Mesoscale Convective Systems to small-sized isolated systems. Further, Galewsky et al. (2023) have linked such differences in convective aggregation to features in δD from IASI. They found that highly aggregated convective systems are characterized by mostly moderate precipitation and less pronounced δD depletion in tropospheric water vapour, whereas unaggregated convection with strong precipitation is associated with more depleted δD signals. This is somewhat contradictory to the result from our cluster analysis as well as from other studies (Risi et al., 2008b; Tremoy et al., 2014; Diekmann et al., 2021a) that mid-tropospheric δD appears relatively more depleted after the passage of Sahelian squall lines, . While squall lines are highly organized systems and, hence, represent aggregated convection, they are associated with deep convective precipitation in the leading edge followed by a trailing stratiform cloud shield with moderate precipitation. Thus, this demonstrates that more refined methods are required to achieve a more comprehensive view on convection and its impact on δD, for instance by combining methods addressing convective aggregation (such as from Galewsky et al. (2023)) and convective depth (e.g., by introducing further and specifically tailored rainfall selection criteria to our clustering method)."*

"l 311: I can see only a few permil drop in fig 7"

The average value for IASI δD in August, as given as difference to the median over all years and as shown in Fig 7, are as follows

| YEAR | δD (‰) |
|------|--------|
| 2015 | 5.96 |
| 2016 | 7.02 |
| 2017 | 14.72 |
| 2018 | -3.97 |
| 2019 | -4.17 |
| 2020 | -11.48 |

l. 311 refers to the drop of δD value in 2017 compared to the values between 2018 – 2020, which lies between -19 and -25 ‰. We will correct the referred δD drop from "-25 ‰" to "down to -25 ‰".

"Fig 7: clarify in the caption what the error bars mean. Is it the standard deviation of all instantaneous values?"

The bars denote the [2.5, 97.5] percentiles of the corresponding distributions. We will add this information into the caption of Fig. 7:

*"The whiskers mark the lower and upper 2.5 percentiles of the underlying distributions."*

"Interpretation of fig 7: To better see the link between rainfall and δD, could a scatter plot of δD vs rainfall anomalies be added?"

We agree that such an analysis can provide interesting insights into the observed amount effect and hence support the interpretation of Fig. 7. Therefore, we will add a corresponding figure showing the correlation of rainfall vs. H2O and rainfall vs. δD using the monthly averaged data for Sahel for IASI and AIRS for the respectively available years (same data as used for Fig. 7):

[Figure]

Here, we observe the discussed anti-correlation between rainfall and δD with decreasing δD as rainfall increases, while H2O is increasing with intensifying rainfall. In this way, it fits well with the discussion of Fig. 7, where corresponding features were observed.

We will add this figure and its discussion accordingly:

*"The comparison of the different years underlines that also an anti-correlation between precipitation amount and δD in the vapour phase can be observed. This becomes particularly evident by contrasting individual H 2 O and δD data points against the corresponding precipitation data, as shown in Fig. 13. While H 2 O increases for intensifying precipitation, we again observe the aforementioned anti-correlated relation between precipitation rates and δD, with higher precipitation rates corresponding to lower δD."*

Interpretation of fig 7: Why are more rainy years more depleted? Is it b ecause there are more rainy events, which are more depleted (fig 6)? Or is it because non rainy events are more depleted? Or because rainy events are more depleted, e.g. because they are more intense? To answer this question, it could be easy to link Fig 7 to fig 6 with a decomposition method: $\Delta\delta D = \Delta r \cdot (\delta D_{rain} - \delta D_{norain}) + r \cdot \Delta\delta D_{rain} + (1 - r) \cdot \Delta\delta D_{norain}$, where $\Delta\delta D$ is the anomaly between high and low rainfall years and r is the fraction of rain samples in the yearly average"

We appreciate this suggestion of utilizing the clustering results for deriving a decomposition method that assesses the contribution of the different factors to the overall δD.

For this purpose, we have investigated the impact of the three factors to the δD anomaly as suggested in the comment:

- $\Delta r \cdot (\delta D_{post-rain} - \delta D_{non-rain})$ as the impact of yearly variations in the fraction of rainfall events

- $r_{post-rain} \cdot \Delta\delta D_{post-rain}$ as the impact of yearly variations in $\delta D_{post-rain}$

- $r_{non-rain} \cdot \Delta\delta D_{non-rain}$ as the impact of yearly variations in $\delta D_{non-rain}$

The following figure shows the results evaluated for IASI, AIRS and TROPOMI for the data used in Fig. 6 (i.e. for June – July of the respective years):

[Figure]

This figure shows that for the considered years the anomalies in $\delta D_{post-rain}$ exhibit high variability in their contribution to the overall δD anomaly, while respective anomalies in rainfall fractions as well as anomalies in $\delta D_{non-rain}$ show only low contribution to $\Delta\delta D$. Since the latter implies that rainfall fractions have been overall stable during the considered time period, this lets us assume that the strong variations in $\delta D_{post-rain}$ are resulting from rainfalls with varying intensity.

Since Fig. 6 (and consequentially also the decomposition results) refer to June – July data and Fig. 7 to August data, their results cannot be linked directly to each other. However, we observe in Fig. 7 that the rainfall peaks have been stronger for the years 2018 – 2020 compared to 2015 – 2017, what matches with the overall results from the decomposition method, where e.g. for IASI the $\delta D_{post-rain}$ anomalies reach minimum values for 2018 and 2019 and are substantially lower compared to 2015 and 2016. This would let us assume that stronger rainfall events account for the negative $\delta D_{post-rain}$ anomalies and, hence, for the drop in δD for 2018 – 2020 as observed in Fig. 7.

This assumption would be supported by the correlation plot of δD vs rain (shown in the response for the previous comment), where stronger rainfall rates go along with decreased δD values.

We added following description and analysis of the decomposition method:

*"By utilizing the results of the precipitation clustering method, we derive a decomposition method that assesses to which extent the non-rain and post-rain clusters may prove useful to explain year-to-year variations in δD as observed in the satellite datasets. For this purpose, we focus on ΔδD as the overall year-to-year anomaly in δD and decompose it by estimating the contribution of the following control factors per year i:*

*ΔδD$_i$ = ΔδD$_{r,i}$ + ΔδD$_{non-rain,i}$ + ΔδD$_{post-rain,i}$ + c*

*with*

$$\Delta \delta D_{r,i} = \left( f_{post-rain,i} - \overline{f}_{post-rain} \right) * \left( \overline{\delta D}_{post-rain} - \overline{\delta D}_{non-rain} \right)$$

$$\Delta \delta D_{non-rain,i} = \overline{f}_{non-rain} * \left( \delta D_{non-rain} - \overline{\delta D}_{non-rain} \right)$$

$$\Delta \delta D_{post-rain,i} = \overline{f}_{post-rain} * \left( \delta D_{post-rain} - \overline{\delta D}_{post-rain} \right)$$

*Here, ΔδD$_r$ represents the impact of year-to-year anomalies in the fraction of post-rain events, f$_{post-rain}$ . Analogously, ΔδD$_{non-rain}$ refers to the contribution to ΔδD due to anomalies in δD$_{non-rain}$ and ΔδD$_{post-rain}$ due to anomalies in δD$_{post-rain}$. f$_{non-rain}$ is the fraction of non-rain events. The variables marked with a bar refer to the averaged value over all considered years, and the variables with the subscript i denote the respective values per year. c refers to the residual anomalies in δD that cannot be addressed by the considered clustering method (e.g. the non-rain and post-rain clusters cover only the edge cases of precipitation rates and not the range in between) and, hence, is not part of this study. Consequentially, the described decomposition method will not provide a complete view on year-to-year anomalies in δD, but will focus on the separate contributions of the described control factors."*

*"From Fig. 10 we can estimate the parameters required for applying the decomposition method for year-to-year anomalies in δD, as described in Sect. 2.2. In particular, this includes the δD values for the individual years and the multi-year average for the non-rain and post-rain clusters as well as the per-year and averaged fractions of non-rain and post-rain events. The δD anomalies resulting from variations in rainfall fraction (ΔδD r ) as well as from variations in δD for non-rain and post-rain are shown per instrument and year in Fig. 11. The results indicate that for the considered years the anomalies in δD of the post-rain situations exhibit high year-to-year variability to the overall δD anomaly, while respective anomalies in rainfall fractions as well in δD non-rain show only minor relative contribution to ΔδD. Since the latter implies that rainfall fraction has been overall stable during the considered time period, this lets us conclude that the strong variations in δD post-rain are resulting from convective events with varying intensity, in terms of e.g. precipitation rates and dry air intrusions. The differences in ΔδD post-rain observed between IASI and AIRS may arise from the different sampling. With the spatial coverage of AIRS being much lower than IASI, it makes the statistical analysis of AIRS more sensitive to outliers, while for IASI the results become smoother. This would explain why for positive ΔδD post-rain values, AIRS is higher than IASI and for negative values it generally appears lower. The overall low contributions of δD non-rain to ΔδD in all three datasets is driven by the much lower fraction of non-rain events compared to post-rain (see Fig. 7)."*

*"Despite differences in the considered time frames, the analysis of Fig. 12 with respect to the decomposition of δD anomalies shown in Fig. 11 provides further insights in this context. Following the discussion of the latter in Sect. 3.3, the decomposition method has revealed that the main driver for year-to-year anomalies in δD are data samples that have experienced significant convective events, i.e. the post-rain events. As the overall fraction of post-rain air masses has been stable over the considered period (see Fig. 11a), the conclusion was made that the variations in δD post-rain may be a result of varying intensity of the convective events. In particular for the years 2018 and 2019, the IASI δD post-rain anomalies reach minimum values and are substantially lower than for 2015 and 2016. Since 2018*

*and 2019 were associated with stronger precipitation peaks than 2015 and 2016 (see Fig. 12), this suggests that stronger precipitation rates account for the negative δD post-rain in Fig. 11 and, hence, for the drop in δD for 2018 – 2020 as observed in Fig. 12. However, these results should be treated with a degree of caution, as Fig. 11 refers to observations from June – July and Fig. 12 from August.”*

"Around l 365: is it possible that the smaller sensitivity of AIRS could be due to the larger impact of the a-priori profile on AIRS than on IASI, i.e. smaller sensitivity?”

Thank you for pointing out this detail. We agree that the observed discrepancies in the {H2O, δD} pair data between IASI and AIRS in low H2O regimes might result from differences in the sensitivity. As is described in Sec. 2.1, the processing of the IASI data considers of a post-processing step that aims at increasing the sensitivity of the {H2O, δD} pairs at dry conditions, what however is not considered for AIRS. We will update this paragraph accordingly.

*"This means that for dry conditions the AIRS data follow significantly less a mixing line if compared to IASI. We assume that this can be explained by differences in the $H_2O$ and δD sensitivities for dry conditions. Typically, retrieval results for δD have much lower sensitivity than for H2O, i.e. the paired {H2O, δD} data product captures well the decrease in $H_2O$ with altitude, but falls short in accurately depicting the decrease in δD (see also Section 7 in Schneider et al. (2016)). The {H2O, δD} harmonization as part of the MUSICA IASI post-processing (see Section 2.1.1) serves to increase the sensitivity of the δD retrieval to dry conditions and thus ensures very similar sensitivities of $H_2O$ and δD, with the effect that the respective IASI {$H_2O$, δD} pairs are distributed close to the mixing line.”*

"l 384: "As result" -> "As a result”

Ok, will be corrected.

"Somewhere: the recent study by [Dahinden et al., 2023] would deserve to be cited.”

Thank you for pointing out this study, which sounds indeed very interesting. We will add a reference to this study in Sec. 5.2 when describing the impact of Saharan air layers to the mid-tropospheric {H2O, δD} pair data:

*"This mixing peaks in the dust loaded Saharan air layer. Respective high δD values for this layer have been reported in several studies and using in-situ measurements from ground and aircraft (Dyroff et al., 2015; González et al., 2016), using data from ground- and space-based remote sensing (Schneider et al., 2015) and using isotopic-enabled modeling (Dahinden et al., 2023).”*

**Literature**

Lafore, J. P., Chapelon, N., Diop, M., Gueye, B., Largeron, Y., Lepape, S., Ndiaye, O., Parker, D. J., Poan, E., Roca, R., Roehrig, R., Taylor, C., and Moncrieff, M.: Deep Convection, chap. 3, pp. 90–129, John Wiley & Sons, Ltd, https://doi.org/https://doi.org/10.1002/9781118391297.ch3, 2017.

**Author Response to the Comments of Referee #2 to the manuscript "Water vapour isotopes over West Africa as observed from space: which processes control tropospheric H2O/HDO pair distributions?" [EGUSPHERE-2024-1613] submitted to Atmospheric Chemistry and Physics.**

We would kindly thank the anonymous referee for providing a review of the manuscript. The individual comments are listed below (shown in red) including our responses (shown in black).

*"This manuscript investigates water vapor isotopes (H2O and HDO) over West Africa, focusing on the West African Monsoon (WAM). Using satellite data from IASI, AIRS, and TROPOMI, the study analyzes moisture pathways and isotopic variations during different monsoon phases. Key findings include an anti-correlation between H2O and δD over the Sahel during monsoon convection, driven by dry air intrusions, while the Guinea Coast shows moist, enriched air without significant δD depletion. The study highlights the value of paired isotopic observations in understanding tropical convection and moisture processes in the region. Overall, this manuscript is well written, and no additional observations or model simulations seem necessary. However, a few minor revisions or updates could help clarify the study and improve the understanding of WAM mechanisms."*

Thank you for the constructive feedback!

**Comments**

*"1. Monsoon retreat phase. The monsoon consists of rainfall systems moving the convection area north and south. However, this study covers the monsoon onset but lacks analysis of the retreat (withdrawal) phase. There could be asymmetry due to land-sea contrasts and seasonal mean state differences."*

This is indeed an interesting research question. In our study, the clustering method applied to identify non- and post-rain events is targeting the monsoon onset phase, as here we expect a clear and sharp transition from the mixing-dominated pre-onset stage to the convection-dominated post-onset stage. In this way, we aimed to characterize the impact due to convection as opposed to the impact due to air mass mixing, with both being identified as substantial factors for controlling mid-tropospheric {H2O, δD} data (see e.g. Diekmann et al. 2021a).
We agree that a potential asymmetry during the retreat may exist and could point towards further control factors, however, this would exceed the actual scope of this study. We will add it as limitation to this study and mention it as future perspective:

*"Further, we have put the focus of the clustering method to the monsoon onset phase, with the aim to characterize the impact due to convection as opposed to the impact due to air mass mixing. Therefore, another interesting question for future studies would be to investigate, whether the clustering method would provide the same results for the monsoon retreat phase or whether asymmetries would appear, e.g. due to different land-sea contrasts during the different seasons and seasonal mean state differences. For instance, Diaconescu et al. (2015) have shown that atmospheric models and reanalyses show significant differences in the performances of reproducing the monsoon onset and retreat. Due to the impact of convection, simulating the monsoon onset remains a key challenge for the models, while the modelled retreat phase is more in line with observational datasets. This points to differences in the main processes controlling the atmospheric state over the Sahel and, for this purpose, our clustering method would be valuable to evaluate to which extent dynamical and microphysical processes change from an isotopic perspective when compared to the onset."*

*"2. Figures showing contrast. Throughout the figures (e.g., regarding non-rain & post-rain periods, Guinea Coast & Sahel), the differences are not clearly visible (except during the monsoon peak month of August) for readers to capture the changes in H2O and δD pairs. The authors may consider*

adding additional figures highlighting these differences to better illustrate which processes or regions are associated with enrichment or depletion as moisture changes."

Thank you for the valuable feedback. We understand that some plots throughout our manuscript contain quite a lot of information and that it may become difficult to grasp the relevant differences in H2O and δD. Therefore, we will add the following figures and the corresponding discussions with the aim to further underline the differences between distributions of H2O and δD:

→ The following figure complements the analysis for Fig. 9 that shows the paired {H2O, δD} distributions of IASI, AIRS and TROPOMI for the months February, May, August and November for the respectively available years. To underline the differences in H2O and δD for the different periods and regions, we have visualized the H2O and δD distributions from Fig. 9 separately as probability density functions. It reflects the features described throughout the analysis of Fig. 9, e.g. that for the Sahel data from IASI and AIRS δD shows a clear drop from winter to summer, while H2O increases. In contrast, over the Guinea Coast we don't observe aconsiderable change in H2O between winter and summer, whereas the summer δD is lower than winter δD.
In this way, this figure further underlines the contrasts between H2O and δD distributions for the different regions and time periods:

[Figure]

with following description:

*"As a next step of cross-comparing the different satellite datasets, we now focus on the distribution shapes of H2O and δD from a qualitative point of view, as shown by the respective probability density functions (PDF) in Fig. 5. We observe that in particular IASI and AIRS show great agreement for both H2O and δD, with the largest discrepancy again for the Sahelian February data. As has been discussed above, IASI reaches significantly lower δD values for dry regimes, which has the effect that the corresponding PDF stretches and flattens compared to AIRS. Further, the TROPOMI PDFs depict overall agreement with IASI and AIRS for H2O, however, reveal discrepancies for δD, e.g. in terms of relative*

*PDF positions between February to May and between February to August. This results from the different characteristics of the TROPOMI data being total column products compared to the mid-tropospheric IASI and AIRS products. To better understand such differences with respect to the underlying control processes (see Section 2.3), this study will shed light on the paired {H2O, δD} distributions of the three instruments."*

→ In order to emphasize the contrasts between H2O and δD distributions, we would like to mention the following figure, that was created based on a comment from the other reviewer (see other review) and serves to further underline contrasts in distributions of H2O and δD. It shows the correlation between rainfall vs. H2O and δD vs. rainfall for IASI and AIRS monthly averages of the August data for all respectively available years. It underlines the different response of H2O and δD to increased rainfall rates, where in particular δD shows decreasing trends for increasing rainfall rates, while H2O shows opposing features. In this way, this figure clearly emphasizes differences in distributions of H2O and δD as response to rainfall:

[Figure]

with the description:

*"This becomes particularly evident by contrasting individual H2O and δD data points against the corresponding precipitation data, as shown in Fig. 13. While H2O increases for intensifying precipitation, we again observe the aforementioned anti-correlated relation between precipitation rates and δD, with higher precipitation rates corresponding to lower δD."*

"3. Satellite dataset uncertainty. The authors mention the previously reported uncertainty in the three satellite datasets, but the reliability of using them specifically for the WAM region is not clearly addressed. Please quantify this uncertainty more explicitly, not only through percentage contours or whisker plots for individual datasets but also by considering the spread between the two datasets (IASI and AIRS). Additionally, the authors could emphasize the benefits of using water isotopes, for instance, by showing more distinct anomalies in δD compared to H2O in those datasets."

Thank you for the constructive suggestion. We have decided to add following figures and their discussion to the manuscript:

→ The following figure shows the scatter between IASI and AIRS for the regions and periods shown in Fig. 9 (i.e. Guinea Coast vs. Sahel, during February, May, August and November, number corresponding to the figure number from the initial manuscript). It demonstrates the overall good agreement between the data from the instruments and reflects differences that have been discussed

in the course of Fig. 9, e.g. deviations in δD for low values, bias in IASI towards higher H2O and towards lower δD values:

[Figure]

[Figure]

with the description:

*"As the isotopic datasets of both IASI and AIRS are vertically resolved with main sensitivity in the free troposphere, we can directly compare their daily H2O and δD distributions for the Sahel and the Guinea Coast after interpolating the data to a common reference grid with 2.5 ∘ degrees grid size (see Fig. 2 and 3). Overall, the two satellites show great agreement for both H2O and δD, with, however, slight deviations for certain regimes. For instance, we observe that IASI depicts a pull to higher H2O ranges for moist conditions compared to AIRS. Further, we observe that for low δD IASI appears more depleted than AIRS, which becomes particularly apparent over the Sahel during February and November. Such differences between the two datasets are expectable due to differences in the respective instrument and retrieval characteristics. For instance, IASI has a higher spectral resolution (0.5 cm−1) compared to AIRS (1 cm−1) and, hence, is more sensitive to atmospheric changes of H2O and δD. Further, the MUSICA IASI*

*{H2O, δD} pair retrieval includes various optimizations to harmonize and increase the sensitivities of H2O and δD, e.g. allowing to better capture changes for dry conditions (see Section 2.1.1)."*

→ In addition, the figure containing the PDFs for H2O and δD from the response to the previous comment serves to further assess the agreement between IASI and AIRS. Despite some differences in their absolute comparison (as shown in the previous figure), the PDFs of H2O and δD reveal that the overall distribution shapes are well-reflected in both datasets for the considered regions and time periods. Probably the largest difference appears for the δD PDFs during February, which however is a result of the observation that IASI covers a larger range in δD compared to AIRS during February, so that the corresponding IASI δD PDF is wider and more shallow than the AIRS δD PDF.

We will add the figures and the corresponding discussions to the relevant sections (see previous comment).

---

## Author Response (AR2)

**Author Response to the Editor Comments to the manuscript "Water vapour isotopes over West Africa as observed from space: which processes control tropospheric H2O/HDO pair distributions?" [EGUSPHERE-2024-1613] submitted to Atmospheric Chemistry and Physics.**

We would kindly thank the Editor Ann Fridlind and the anonynomus referees for providing a review of the revised manuscript. The individual comments are listed below (shown in red) including our responses (shown in black) and the changes made in the manuscript (shown in black and italics).

**Comments**

"Section "Author contribution": Please use initials for the authors' names."

Thank you for this note. We have updated the section correspondingly.

"Please bring the abstract into compliance with guidelines for authors:

https://www.atmospheric-chemistry-and-physics.net/policies/guidelines_for_authors.html

Then I will be glad to accept this manuscript for publication."

In alignment with the referred guideline, we have shortened the abstract to 250 words:

*"The West African Monsoon (WAM) is crucial for rainfall in West Africa, impacting socio-economic conditions. Its complexity arises from interactions between large-scale circulation, convective dynamics, and microphysical processes, making it challenging to disentangle individual contributions to the hydrological cycle.*
*Recent advances in retrieving the isotopic composition of tropospheric water vapour from space promote the paired analysis of $H_2O$ and HDO to study atmospheric moisture pathways and processes. Using data from satellite instruments IASI, AIRS and TROPOMI, along the IMERG precipitation product, we analyse the variability of $H_2O$ and HDO (given as $\delta D$) over West Africa at convective and seasonal scales. Key findings include:*

*(1) Monsoon convection over the Sahel induces an anti-correlation between $H_2O$ and $\delta D$ in the mid-troposphere. This is due to dry intrusions from the Saharan upper troposphere into Sahelian squall lines, fostering rain evaporation and mid-tropospheric $\delta D$ depletion.*
*(2) Over the Guinea Coast, convective precipitation is associated with moist and enriched signals, with surface evaporation from the Tropical Atlantic reducing rain evaporation and $\delta D$ depletion.*
*(3) During the Sahelian monsoon peak, an anti-correlation between precipitation and $\delta D$ forms year-to-year, indicating the amount effect in tropospheric water vapour.*
*(4) In the Sahelian winter, when precipitation is minimal, {$H_2O$, $\delta D$} signals point to mixing of dry air masses from different origins.*

*This study is the first to apply comprehensive isotopic datasets from IASI, TROPOMI and AIRS to the WAM, demonstrating the utility of satellite-based {$H_2O$, $\delta D$}-pairs in detecting impacts of microphysical and dynamical processes on water vapour isotopic composition."*